# Policy and market forces delay real estate price declines on the US coast

Dylan E. McNamara [1,2,7], Martin D. Smith [3,4,7] ✉, Zachary Williams[1,3], Sathya Gopalakrishnan[5] & Craig E. Landry[6]

Despite increasing risks from sea-level rise (SLR) and storms, US coastal communities continue to attract relatively high-income residents, and coastal property values continue to rise. To understand this seeming paradox and explore policy responses, we develop the Coastal Home Ownership Model (C-HOM) and analyze the long-term evolution of coastal real estate markets. C-HOM incorporates changing physical attributes of the coast, economic values of these attributes, and dynamic risks associated with storms and flooding. Resident owners, renters, and non-resident investors jointly determine coastal property values and the policy choices that influence the physical evolution of the coast. In the coupled system, we find that subsidies for coastal management, such as beach nourishment, tax advantages for high-income property owners, and stable or increasing property values outside the coastal zone all dampen the effects of SLR on coastal property values. The effects, however, are temporary and only delay precipitous declines as total inundation approaches. By removing subsidies, prices would more accurately reflect risks from SLR but also trigger more coastal gentrification, as relatively high-income owners enter the market and self-finance nourishment. Our results suggest a policy tradeoff between slowing demographic transitions in coastal communities and allowing property markets to adjust smoothly to risks from climate change.

Many coastal communities exist precariously close to mean sea level[1] but continue to attract residents and investment. Sea-level rise (SLR) will eventually submerge vast swaths of low-lying coastal landscapes, and shoreline management that maintains beaches and dunes will be unable to keep pace with inundation[2]. Some places are likely to be uninhabitable within this century[3,4]. A number of coastal hazard risks are already increasing due to SLR[5], and millions of continental US households are at risk of inundation from SLR by 2100[6]. Despite increasing risks from SLR and storms, coastal real estate sells at a premium (Fig. 1). In the US, coastal real estate has been appreciating faster than non-coastal real estate[7], and coastal residents have higher incomes than non-coastal residents[8].

Growing risks in coastal areas and high real estate prices seem paradoxical. If inundation is inevitable in the long run, why do high prices for coastal real estate persist with a demographic trend toward high-income owners? It appears as if people are racing to get in the way of climate change. This behavior is consistent with empirical findings that property prices do not fully reflect the risks from SLR and flooding[9–11], and properties vulnerable to flooding have inflated prices relative to market fundamentals[12]. Nevertheless, other empirical

[1]Department of Physics and Physical Oceanography, UNC Wilmington, Wilmington, NC 28403, USA. [2]Center for Marine Science, UNC Wilmington, Wilmington, NC 28403, USA. [3]Nicholas School of the Environment, Duke University, Durham, NC 27708, USA. [4]Department of Economics, Duke University, Durham, NC 27708, USA. [5]Department of Agricultural, Environmental, and Development Economics, The Ohio State University, Columbus, OH 43210, USA. [6]Department of Agricultural and Applied Economics, University of Georgia, Athens, GA 30602, USA. [7]These authors contributed equally: Dylan E. McNamara, Martin D. Smith. ✉e-mail: martin.smith@duke.edu

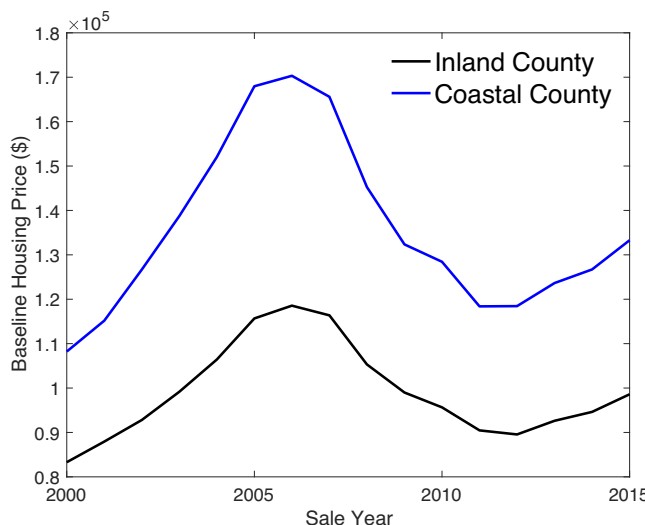

**Fig. 1 | Coastal and inland property values.** Quality-adjusted baseline housing values are reflected in estimated county-year fixed effects for coastal (blue) and inland (black) counties in the United States. Each county-year fixed effect comes from a first-stage hedonic price function that estimates the natural log of property sales price, controlling for property characteristics, including square footage, bedrooms, bathrooms, lot size, and construction type, using 23,184,659 observations of property sales in the United States between 1989 and 2016. The first-stage hedonic regression was run in Stata/SE 18.0.

evidence shows that properties vulnerable to SLR are discounted relative to comparable properties that are not vulnerable, suggesting that some buyers take climate risks into consideration[13]. Similarly, housing markets capitalize storm risks in the short run, although price discounts dissipate within several years of a storm[14,15]. The Chief Economist of the National Association of Realtors, Lawrence Yun, succinctly summarized the situation, "Consumers are clearly mindful that these climate change impacts could be within the window of a 30-year mortgage, but their current behavior still implies that to have a view of the ocean is more desirable"[16]. Consonant with a strong desire for coastal living, after a major storm event, communities rebuild with even larger homes[17], and the costs of adaptation measures, such as beach nourishment projects that periodically rebuild eroded beaches and dunes, often are easily justified by the avoided property value losses[18].

To understand the mechanisms driving these empirical phenomena and what to expect in coastal communities as SLR progresses, it is necessary to model how real estate markets interact with the physical coastal system over a long time horizon. Empirically isolating the SLR signal in property prices is challenging because it may be small relative to other market fluctuations and drivers of market fundamentals. Moreover, the worst anticipated SLR effects may occur toward the end of a 30-year mortgage and thus be heavily discounted in current decisions. The range of experience of SLR and storm risk captured in empirical studies may not include the full range of possibilities under future scenarios. For example, if risks respond nonlinearly as SLR progresses beyond what has been observed in the past, only modeling studies are capable of exploring the implications. To consider large and long-term effects of SLR, a model is needed that links real estate markets, coastal amenities, coastal hazards, and policies that respond to coastal change.

This work contributes to advancing the growing literature on coupled human and natural systems. Humans are constantly changing the natural environment surrounding them, and understanding dynamic feedbacks between human behavior and natural systems often requires more than just superimposing an economic model on the physical or biological system[19]. Applications of coupled modeling

of dynamic human-natural feedbacks dates back at least to the 1960s when bioeconomic models were used to study the human and natural components of fisheries[20,21]. The literature expanded dramatically when researchers began using spatially explicit data to study land use and land cover change, urbanization patterns, and to evaluate conservation interventions[22–27]. Progress in understanding coupled systems adds complexity by modeling nonlinear feedbacks between physical processes and human responses across space and time[19,28].

In coastal systems, the evolution of the coastal-economic zone cannot be understood with methods in economics or coastal modeling alone; rather, it depends on complex interactions between physical coastal systems and economic behavior[29,30]. In these systems, incorporating relatively simple models of human behavior with a detailed geophysical model of coastal evolution[31,32] and coupling simplified dynamics of coastal change with detailed economic decision-making[33,34] can generate new insights and emergent patterns in the coupled system[35]. Adding complexity in any one dimension can reveal system characteristics that may not be consistent with simpler models. To add to this literature, it is necessary to endogenize real estate values and demographic changes as functions of SLR risk in a model that also includes model couplings and features from this previous work, namely beach erosion, storm risk, the effects of beach width on property value, and local public finance decisions to rebuild beaches.

In the absence of modeling of a coupled human-natural system, researchers can also misinterpret empirical results and potentially draw the wrong policy implications[36–38]. Even in simple models of coupled systems, state variables behave in non-intuitive ways such as being positively correlated over some time intervals and negatively correlated over others[21]. As such, there is a growing need to use modeling to evaluate the reliability and plausibility of empirical evidence for causal claims and to elucidate potential mechanisms for surprising empirical findings[37,38]. The use of coupled systems modeling to inform empirical specifications can also lead to substantially different estimates, such as a value of beach width that is more than double the estimate that ignores the coupling[39].

Coastal communities along the US East and Gulf coasts, which are highly developed, densely populated, and have home prices that are higher than the national average, provide motivation for modeling the coupled system. Much of the real estate in this region is in highly erodible sandy coastal areas, including oceanfront and nearshore (non-oceanfront) properties that are built on low-lying barrier islands (Fig. S1). Such areas are vulnerable to a myriad of climate-related impacts, including damage from wind and storm surge due to increased frequency and intensity of storms as well as impacts from SLR, which can intensify surge-related flood risk, increase erodibility of the shoreline, increase the frequency of sunny-day flooding, threaten groundwater, and eventually lead to total inundation. Barrier island communities constitute a natural boundary to define coastal housing markets and to measure the extent of the impact of local adaptation measures. On developed coastal barrier islands, oceanfront and near shore housing markets are often fully developed and changes in housing supply tend to reflect damage to property and building back homes after hurricanes[17]. In a recent empirical study of coastal development on the southern barrier islands in North Carolina, only 8% of parcels were newly developed between 1993 and 2013[40], with a smaller percentage of developable oceanfront parcels. Holding housing supply fixed in modeling enables an examination of demand-driven market dynamics without loss of generality.

Here, we show that subsidies for coastal management, such as beach nourishment, tax advantages for high-income property owners, and stable or increasing property values outside the coastal zone all dampen the impacts of SLR on coastal property markets. The effects are transitory and only delay precipitous price declines as total inundation approaches. Prices would more accurately reflect SLR risks without subsidies, but coastal gentrification

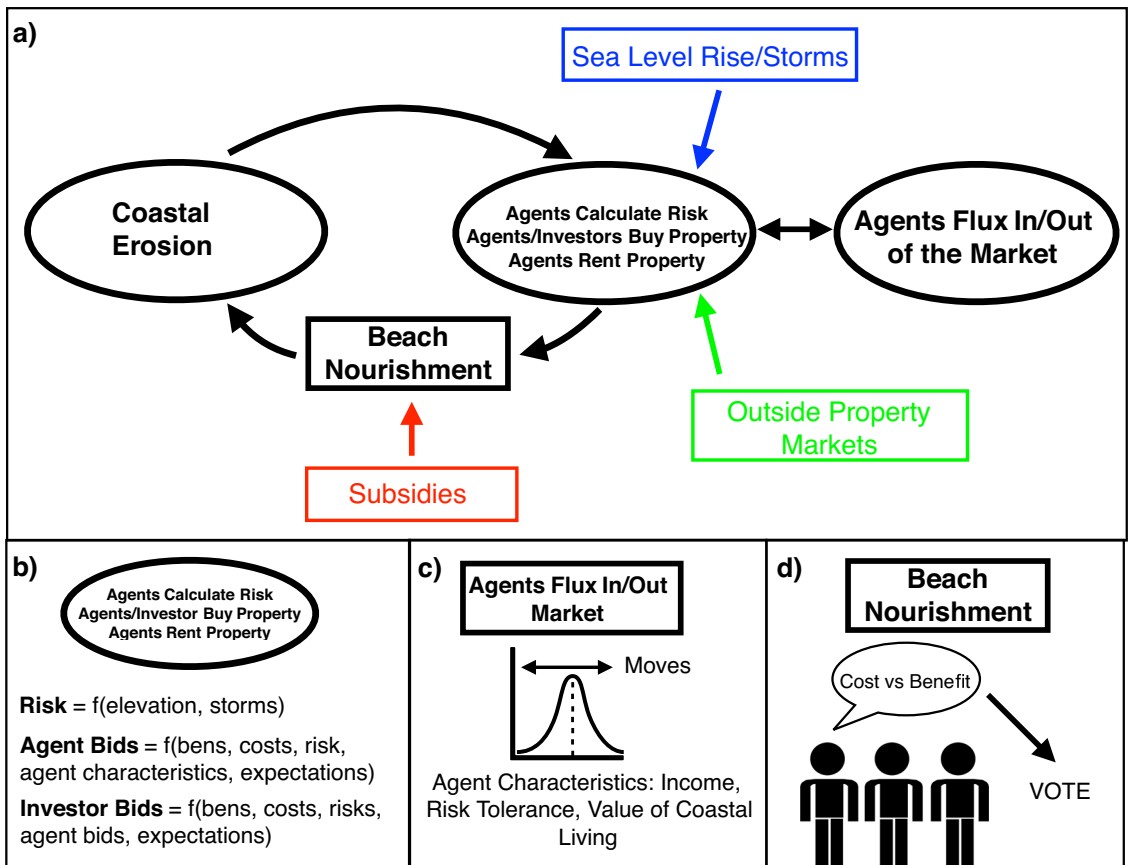

**Fig. 2 | Stylized modeling framework.** Depicts internal model dynamics (black) for which agent actions and outcomes are endogenously determined and external forcings (red, green, and blue) that exogenously influence actions and outcomes for the coupled human-natural system (**a**). The remaining panels provide details on human decisions, including risk assessments and property valuations (**b**), owner agent flux (**c**), and beach nourishment decisions (**d**).

would accelerate, as relatively high-income owners enter the market and self-finance nourishment. Our results suggest a policy tradeoff between slowing demographic transitions in coastal communities and allowing property markets to adjust smoothly to risks from climate change.

We study the evolution of coastal real estate markets, incomes of residents, and shoreline management decisions in response to SLR over a 150-year time horizon by developing the Coastal Home Ownership Model (C-HOM). We define income in relative terms based on US marginal income tax rates that range from a low of 10 percent to a high of 37 percent. The 150-year simulation length allows us to consider longer horizons than a typical 30-year mortgage; run the model for 50 years without SLR as an initial period to understand internal mechanisms in the model; and evaluate large, long-term effects of SLR and a changing storm climate (over the subsequent 100 years). Coastal property markets and physical processes are a coupled system with feedbacks between agent actions (e.g., buying and selling property, voting, and beach management) and the physical system (e.g., erosion and beach width) (Fig. 2); the human system affects the natural system, and vice versa. The model also considers exogenous forces that influence internal dynamics, including SLR, storm risk, and the influence of competing property markets (Fig. 2). We take as given that there is enough SLR locked-in to cause widespread coastal inundation in the long run, and we use the model to probe previously unanswered questions. What mechanisms allow a coastal community to delay the inevitable collapse of a coastal real estate market? In the long run, how might interventions to defend the coast influence demographic changes?

## Results

Four model scenarios explore how internal and external dynamics affect the coupled coastal system (Fig. 2). Model parameters are in Supplementary Table S1. The first scenario is a baseline simulation used to evaluate the impacts of changing external conditions or policy, i.e., the counterfactual to which other scenarios are compared. In all scenarios, the physical environment has a constant rate of shoreline erosion, constant storm probability, and one meter of SLR over a 100-year period. In the baseline scenario, 90% of beach nourishment costs are subsidized, and outside markets are held constant, meaning that, after accounting for inflation, nationwide housing markets are not appreciating. This effectively means that in the baseline scenario, returns on investment in real estate are based on the flow of rental income, i.e., associated housing services and owners' Willingness to Pay (WTP) for coastal amenities, and there are no arbitrage opportunities associated with coastal or non-coastal property. The 90% subsidy accounts for the combined effect of state and federal subsidies as well as hotel and other tourism-related taxes that are not embedded in local property tax rates.

In the second scenario, the beach nourishment subsidy is reduced to 50%, while outside markets remain constant. In the third scenario, the nourishment subsidy is returned to 90%, but outside markets appreciate after 50 years, such that real inland housing prices double over the next 50 years and then remain at that level over the last 50 years of the simulation. In the final scenario, nourishment is subsidized at 90%; outside property markets remain constant 50 years after the onset of SLR, but outside values depreciate over the subsequent 50-year period, declining to 10% of their initial value by the end of the simulation. Detailed

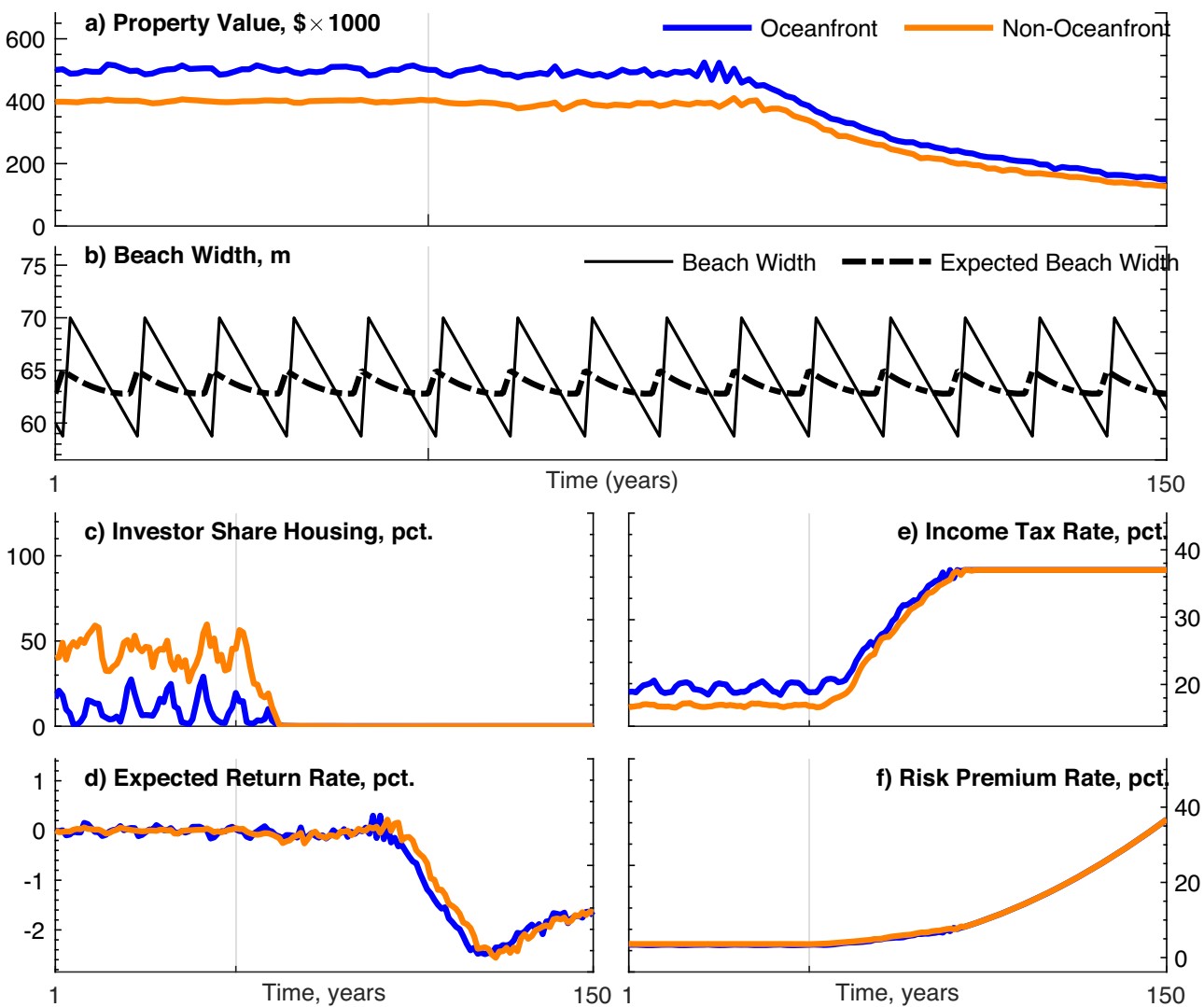

**Fig. 3 | Baseline model scenario.** The baseline case assumes that SLR begins at $t = 50$ (marked by a vertical gray line) and continues throughout the remaining periods. Beach nourishment is subsidized at 90% of the cost. Outside property prices are constant, and agent flux is turned on (relatively high-income agents are able to enter to exploit arbitrage opportunities with outside markets). Depicts model simulation outcomes for oceanfront properties (blue), non-oceanfront properties (orange), beach width (solid black), and expected beach width (dashed black). Outcomes include property value (**a**), beach width (**b**), share of housing owned by investors (**c**), expected rate of return on coastal real estate (**d**), marginal income tax rate of property owners (**e**), and risk premium for coastal real estate (**f**).

justifications for the scenarios are in the Supplemental Materials along with sensitivity analysis.

In each scenario, the initial barrier height, i.e. elevation of the barrier island, relative to sea level is 1 m for the first 50 years. SLR begins at year 50 and rises 1 m over the remaining 100 years. By model year 150, the barrier height relative to mean sea level is 0 m, i.e., the threshold of total inundation. The initial 50-year period characterizes the system without SLR.

In the baseline, over the first 50 years, the housing market is in dynamic equilibrium with the physical environment ($MSL_t = \overline{MSL}$ for $t \leq 50$) and with external economic conditions ($P^e$). The oceanfront (blue) and non-oceanfront (orange) property values are approximately equal to outside market prices (Fig. 3a). Small amplitude price fluctuations reflect changes in the shoreline position due to erosion and subsequent nourishment (Fig. 3b) and translate into small fluctuations in expected returns (Fig. 3d). Fluctuations are more pronounced in oceanfront property values because beach width has more effect on property value for oceanfront homes. Investors own a portion of each market segment, and owner agents have a relatively low median income tax rate (Fig. 3c, e).

When SLR begins in year 50, property values decline modestly, and socioeconomic characteristics of the community change. A substantial decline in property value does not occur immediately, taking decades past the onset of SLR to manifest, which results from an initially mild impact of SLR on housing risk (Fig. 3f). Increased risk combined with stable outside property markets drives an influx of relatively high-income owner agents whose bids reflect amenity values in outside markets and consequently maintain property values despite SLR (Fig. 3e). High-income agents have higher bid prices relative to institutional investors because they have higher marginal income tax rates compared to the corporate tax rate. Eventually, as property values begin to decline more substantially, the investor market share declines to zero (Fig. 3c). This crowds out investors and lower income renters. The collective impact of increasingly high-income agents entering the market is to temporarily dampen the property value decline. Indeed, property values decline immediately if relatively high-income agents are not allowed to enter. To test this mechanism, we analyze the case in which agents are not allowed to flux in to exploit arbitrage opportunities with outside markets. Here we see that SLR translates into an immediate decline in property value (Supplementary

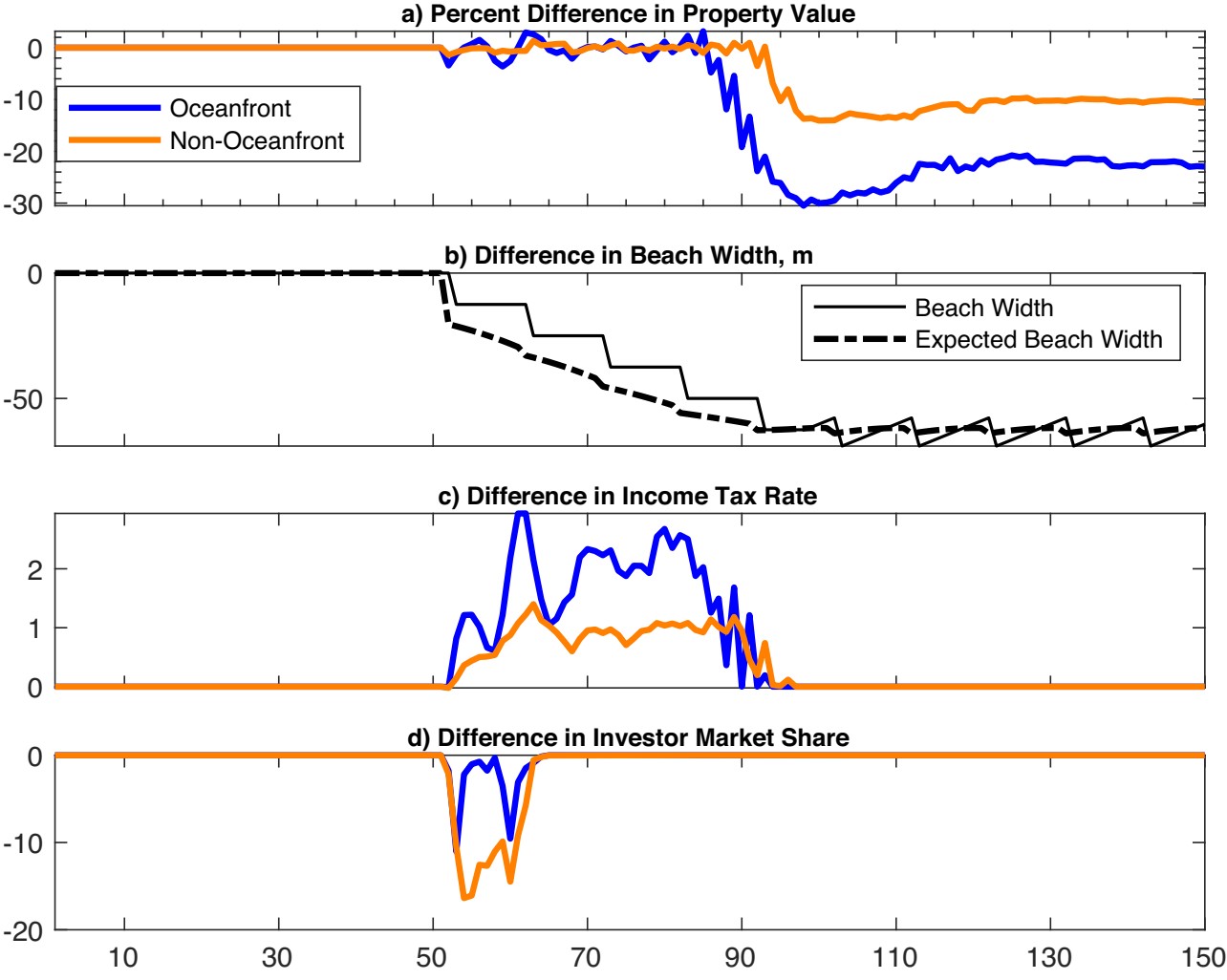

**Fig. 4 | Numerical experiment that cuts the nourishment subsidy.** Results displayed are model simulation differences between the baseline model and a scenario in which the subsidy rate is cut to 50% at the onset of SLR (at *t* = 50). Depicts outcomes for oceanfront properties (blue), non-oceanfront properties (orange), beach width (solid black), and expected beach width (dashed black). Outcomes include percent change in property value (**a**), change in beach width (**b**), change in marginal income tax rate of property owners (**c**), and change in the share of housing owned by investors (**d**).

Fig. S2a). This decline is not associated with a decrease in beach width, as communities continue to nourish at the same frequency with the nourishment subsidy intact (Supplementary Fig. S2b). The corresponding difference in marginal tax rate compared to the baseline declines, reflecting the lack of relatively high-income owners fluxing into the system (Supplementary Fig. S2c). As in the baseline case, investors mostly leave the market as SLR worsens, but some share of investors remains in the oceanfront market (Supplementary Fig. S2d). This reflects the tax advantage of investors that allows them to capture some of the nourishment subsidy and capitalize on high willingness to pay for oceanfront living despite rising seas.

Approximately 60 years after the onset of SLR, property values begin to decline rapidly (Fig. 3a). There is no longer a draw for high-income agents because the maximum income tax rate was saturated at the end of the previous phase, and there is no compensatory mechanism to stabilize or drive prices back up. With risks continuing to increase (Fig. 3f), property values continue on a downward trajectory. Throughout the baseline, the 90% nourishment subsidy leads to a wide beach being maintained throughout the 100 years of SLR with nourishment frequency remaining stable as relativel high-income agents enter the system (Fig. 3b). To these agents, the small increase in property tax is preferred to the amenity value lost with less frequent nourishment.

The baseline demonstrates that property value does not immediately reflect SLR risks. Instead, risk, investment, and income interact to obscure the signal of SLR in property value. Even in the case where outside markets are constant, when the SLR signal begins to affect prices, arbitrage opportunities with outside markets help to maintain high prices as high-income buyers enter the market.

When beach nourishment subsidies are dramatically reduced (from 90% to 50%) in year 50, four important changes occur (Fig. 4). First, less nourishment subsidy initially leads to more volatility in property values and then triggers a precipitous decline after several decades (Fig. 4a). The effect is more pronounced for oceanfront than for non-oceanfront properties. Second, since residents must self-finance a greater portion of the cost of nourishment, mean beach width immediately declines without the subsidy (Fig. 4b). Over several decades, mean beach width levels off and nourishment resumes (Fig. 4b). Third, the median owner income tax rate increases (both oceanfront and non-oceanfront) after reducing the nourishment subsidy (Fig. 4c). This reflects a process by which lower average beach width slightly reduces property values, but these declines create arbitrage opportunities that draw in relatively high-income agents with higher marginal tax rates. Although this process occurs in the baseline, the higher income agents are drawn in sooner with lower nourishment subsidies. Fourth, the investors own substantially less of the housing

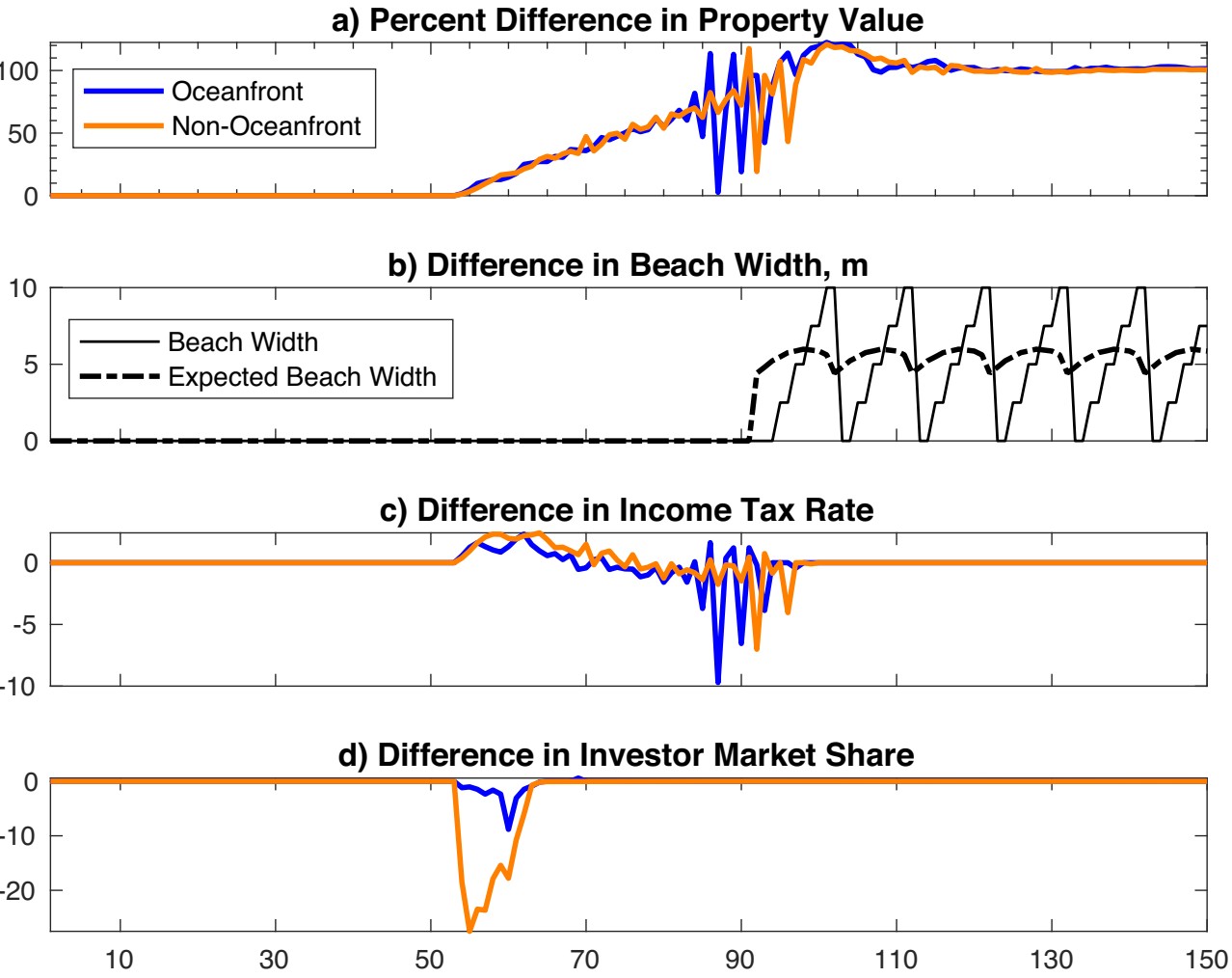

**Fig. 5 | Numerical experiment in which outside markets appreciate.** Results displayed are model simulation differences between the baseline model and the scenario in which outside markets appreciate. SLR begins at $t = 50$. Depicts outcomes for oceanfront properties (blue), non-oceanfront properties (orange), beach width (solid black), and expected beach width (dashed black). Outcomes include percent change in property value (**a**), change in beach width (**b**), change in marginal income tax rate of property owners (**c**), and change in the share of housing owned by investors (**d**).

market immediately after the onset of SLR (Fig. 4d). As relatively high-income agents enter to maintain property values relative to outside markets, they drive out investors and lower income renters in the process.

When prices in outside markets rise, three important differences emerge compared to the baseline. First, coastal real estate prices appreciate well after the onset of SLR and, after 100 years, remain above baseline prices even as barrier elevation approaches zero (Fig. 5a). Second, there is a delayed effect on beach width (Fig. 5b). Initially after SLR begins, beach width is unchanged relative to the baseline because the same frequency of nourishment persists. As housing prices rise and the associated value of beach width increases, however, nourishment and associated beach width increase relative to the baseline. Third, rising prices drive investors out of the market quickly as relatively high-income owners enter sooner (Fig. 5c, d).

We also model the case in which the subsidy is reduced and outside markets rise. We find that the effects of rising outside markets offsets the removal of subsidies in the long run. In this scenario, prices continue to increase in the presence of SLR even as mean beach width declines significantly (Supplementary Fig. S3a, b). Initially, removing the subsidy decreases nourishment, but as property value rises, self-financing of nourishment is sufficiently valuable, and the cycle of nourishment and associated mean beach width resembles the baseline

scenario (Supplementary Fig. S3b). This is made possible by the more rapid influx of high-income owners (relative to the baseline) (Supplementary Fig. S3c), and correspondingly investors are driven out of the market faster relative to the baseline (Supplementary Fig. S3d).

To explore further the influence of outside markets, we compare a world with rising outside markets and SLR to a world with constant outside markets and no SLR. In this case, rising outside markets initially contribute to higher prices despite SLR (compared to constant outside markets and no SLR), but eventually SLR dominates the appreciation in outside markets and property values plummet (Supplementary Fig. S4a). Initially, nourishment is the same in both worlds. However, in the world with rising outside markets, the influx of relatively high-income agents and higher prices eventually justify even more frequent nourishment and wider beaches (Supplementary Fig. S4b). Moreover, investors are completely crowded out, and the owners are permanently higher income compared to the world with constant markets and no SLR (Supplementary Fig. S4c, d).

Lastly, if outside markets decline, the influx of high-income agents can be reversed (Fig. 6). We simulate the decline in outside markets beginning 50 years after the onset of SLR. Initially, nourishment continues as in the baseline, expected returns fluctuate around zero, and the influx of relatively high-income agents crowd out investors and maintain property values despite modest increases in risk from SLR

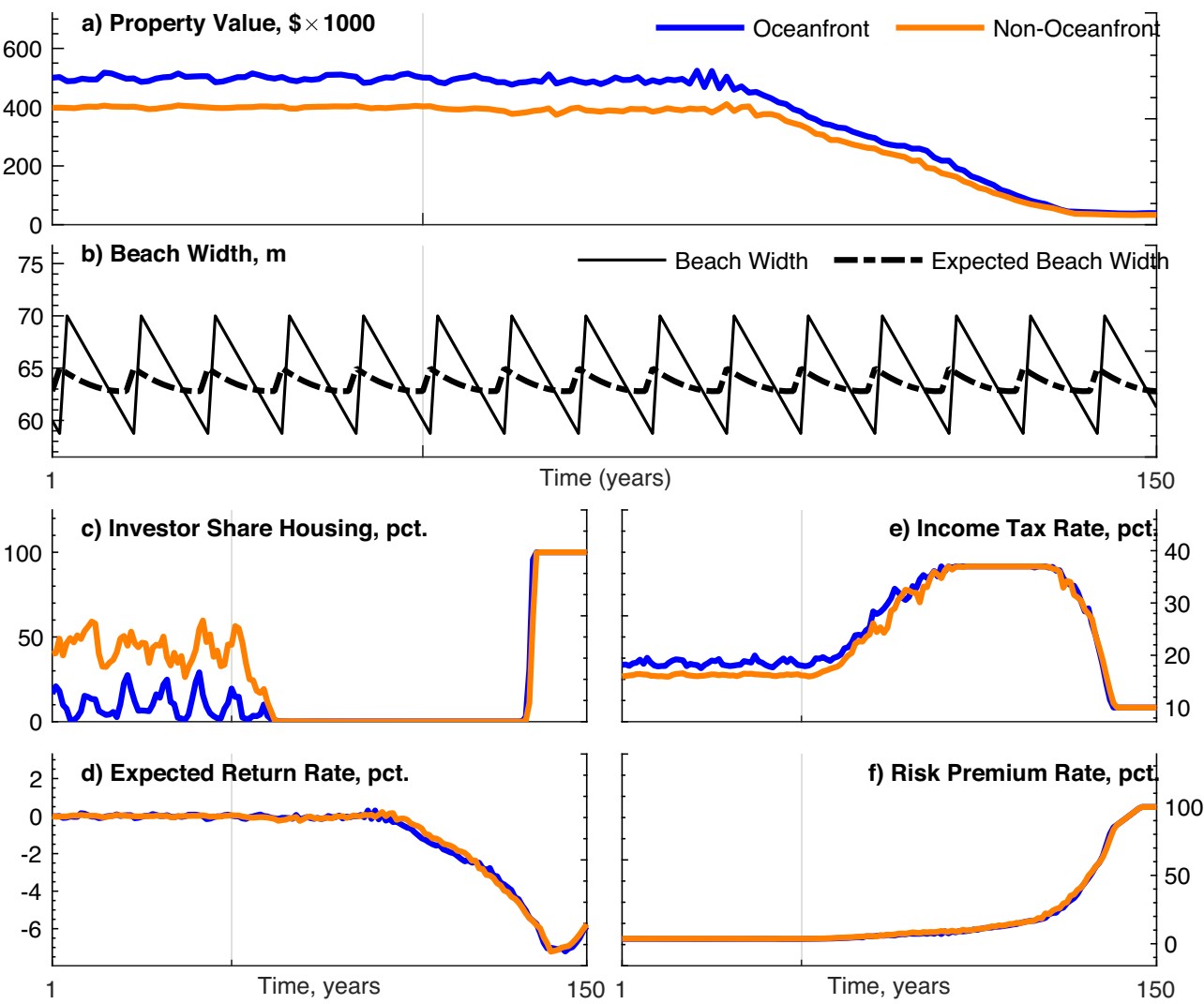

**Fig. 6 | Outside markets decline.** Results displayed are model simulations in which SLR begins in $t = 50$ (marked by a vertical gray line), the baseline scenario with constant outside markets spans the first 50 years after the onset of SLR, and then outside markets decline to 10% of their initial levels over the following 50 years. Depicts outcomes for oceanfront properties (blue), non-oceanfront properties (orange), beach width (solid black), and expected beach width (dashed black). Outcomes include property value (**a**), beach width (**b**), share of housing owned by investors (**c**), expected rate of return on coastal real estate (**d**), marginal income tax rate of property owners (**e**), and risk premium for coastal real estate (**f**).

(Fig. 6a). When outside markets decline, however, expected returns and property values plummet sharply and risk continues to increase. This occurs despite continued nourishment and maintenance of actual and expected beach width (Fig. 6b). Eventually (after roughly 30 years), relatively high-income owners leave the market, investors buy all of the real estate, and rent it to lower-income occupants (Fig. 6c, e).

## Discussion

Coastal real estate markets along the US East and Gulf Coasts are tightly coupled to the physical environment[31,33]. SLR and associated inundation will eventually alter this coupling, and an inevitable major transformation of these markets will ensue. By modeling coastal real estate markets that are coupled to the physical environment, exogenous climate forcing, and the effects of real estate markets in other locations, C-HOM sheds light on potential trajectories associated with this transformation as communities approach the point of inundation. We add to a growing body of literature modeling coupled human-natural systems[19,28,38] and applying a coupled systems framework to landscape change[23,41]. In coupled systems literature that models beach erosion dynamics and coastal real estate, we build on existing models

that couple beach nourishment to property markets[33,39], incorporate local public finance decisions[42], and parameterize storm risk in the coupled system[29]. We include all of these model features and add how SLR drives local real estate market responses and demographic changes. We find that several features of the coupled human-natural system dampen environmental signals and delay the full capitalization of risk from SLR. This helps to explain the seeming paradox that coastal property values continue to rise as climate-related risks increase.

In the present and near future, coastal communities attract wealthier residents despite growing risks from climate change. This demographic shift can be thought of as a form of coastal gentrification[43,44]. Natural amenities that tend to persist have an anchoring effect on high income distributions, and the persistence of coastal amenities partly explains why coastal cities experienced less out-migration into the suburbs in the mid-20th century relative to non-coastal cities[45]. In our model, willingness to pay for coastal living is persistent and can be bolstered by policy interventions such as sub-sidizing beach nourishment. This persistence gentrifies the coast and leads to a long-run anchoring effect. Only when competing property markets decline and the coastal market becomes inflated relative to

them do we see the anchoring effect break down and reverse the course of coastal gentrification.

We specifically highlight three mechanisms that contribute to the persistence of high property values in coastal areas vulnerable to SLR. First, high-income owners are tax advantaged over low-income owners and institutional investors because higher marginal tax rates increase the benefits of the mortgage deduction. This acts as a subsidy and partly explains increasing rates of home ownership in the 1970s[46]. The implicit subsidy to higher-income households is a mechanism driving agents to bid up desirable real estate. To the extent that people with higher income have a higher willingness to pay for coastal amenities, this mechanism puts short-run upward pressure on coastal property values regardless of the expected long-run future state of the system.

Second, tax policy works in concert with outside markets. If coastal prices adjust downward, high-income agents enter the market to exploit arbitrage opportunities with the broader market for desirable real estate, i.e. non-coastal real estate with high amenity values. This props up prices, and ultimately the entry of relatively high-income agents dampens the climate signal. Essentially, the onset of SLR triggers an initial influx of relatively high-income agents that delays the downward adjustment in prices. The rate at which agents enter is thus crucial but not well studied. The extent to which tax policy is reinforcing also depends on whether coastal properties are second homes not subject to the home mortgage deduction. This suggests more empirical work is needed to understand coastal migration and the detailed demographics of ownership. Outside markets also dampen climate signals when they appreciate. In principle, one could isolate a discount for SLR from a background rising trend. In practice, a rising trend increases the tax base, justifies more property enhancing investment, and contributes to sustained high property value despite climate risks. Rising outside markets eventually trigger more frequent beach nourishment despite property elevations approaching zero (Fig. 5b).

Third, policies that artificially increase the value of property (e.g., a beach nourishment subsidy) delay the downward adjustment of prices in response to SLR. With the majority of projects implemented in the US by the Army Corps of Engineers, records maintained by the Program for the Study of Developed Shorelines indicate that cumulative federal expenditures on beach nourishment projects have exceeded 12 billion USD in the past century (in 2022 dollars; https://beachno.wcu.edu/). Although nourishment often passes a benefit-cost test[18,39], previous studies do not consider the effects of nourishment subsidies on stimulating housing demand in areas vulnerable to SLR. By modeling the coupled human-natural system, C-HOM provides a foundation for comparing business-as-usual with potentially more efficient approaches to coastal management and the equity implications of different policy approaches. For example, if subsidies are reduced, prices will adjust downward to climate risk more rapidly. Subsidies also delay associated demographic shifts. Lower subsidization leads to more rapid coastal gentrification but permits greater capitalization of flood risk in coastal property values. Thus, removing subsidies for beach management can improve market efficiency (in that prices reflect risk), but can impede equity (in that lower income households are pushed out).

We raise questions for the empirical literature on the causal impact of climate risk, which is an important role for modeling coupled human-natural systems[38]. A number of studies find empirically that property values fail to capitalize climate risks fully[9,10,12], but the three mechanisms that we identify suggest that these effects may be short-lived. Although preference and/or informational heterogeneity can still play a role in the short run[10,12], information provision alone will not adjust markets to account for long-run risks. The presence of subsidies, tax policy that privileges high-income owners, and escalating outside markets all contribute to sustaining high values of coastal real estate.

The subsidy process resembles disasters studied in ref. 47, which contrasts the effect of tornadoes on out-migration with the effect of flooding on in-migration in the 1920s and 1930s. Provision of public infrastructure, such as sea walls, partly accounts for less out-migration from flood-prone areas. Similarly, our model shows that public funding for nourishment discourages out-migration. Subsidy effects could be even stronger if income inequality continues to rise in the broader US economy. Because lower income residents remain in the community longer with higher subsidies, there is a potential tradeoff between coastal management that discourages gentrification and management that allows prices to reflect long-run risks. Maintaining the historic socioeconomic composition of coastal communities may come at the expense of inducing markets to mask the SLR signal and adjust more slowly to the long-run inevitable outcome. This tradeoff is particularly poignant in light of recent findings that lower-income households face greater risks of losing home equity due to climate-related flooding[11].

It is also possible for market forces to reverse the process of coastal gentrification. With SLR, maintaining a coast populated with high-income owner occupants and high property values is predicated on stable or rising outside markets. If these markets decline (e.g., outside property values adjust downward in response to climate risks, macroeconomic shocks, or tax reform), the result is an influx of lower income renters (Fig. 6a). Lower income residents reinforce the downward pressure on prices from SLR, raising the possibility that climate processes give rise to low-income homogeneous coastal communities. The exit of relatively high-income agents is consistent with climate gentrification in which wealthier residents relocate to less vulnerable areas, leaving less wealthy residents in climate vulnerable areas[48]. Our results show that the same community can experience both coastal gentrification and climate gentrification, occurring during distinct periods of time. Whether and when these phenomena occur hinges on policy interventions and the time paths of changes in the physical system and competing property markets. In the long run-whether the coastal environment becomes low-income homogeneous, is populated by wealthy owner occupants, or is something in the middle-a precipitous decline in the housing market prior to complete inundation may not be desirable. It may be possible that policies can arrest the sharpness of decline and smooth the transition toward inundation.

Our findings raise questions about how public funds could be spent differently to promote adaptation. We find that subsidies for beach nourishment fail to promote adaptation in the long run and could be maladaptive, as suggested elsewhere[29,49], and at best, subsidies delay downward adjustments in real estate markets and demographic shifts. Alternatively, public funds could be spent on more deliberate ways to adaptive the coast to a future with a higher mean sea level. For example, managed retreat aims to smooth the transition to uninhabitable more deliberately[50,51]. One specific approach worthy of further investigation is buyouts with rentbacks, which potentially maintains community composition without inflating property values and without encouraging further development in vulnerable coastal areas[52]. With this policy, local owners are bought out and can remain in their homes temporarily by renting them back, which allows flexibility in retreat from areas facing inundation in the long run without requiring families to shoulder long-run risk. Although reallocating federal spending on beach nourishment would not be enough to do large-scale buyouts of properties in barrier island communities-federal spending on nourishment in the US was USD 229 million in 2022 (https://beachno.wcu.edu/)-it potentially could initiate a buyouts with rentbacks fund with subsequent funding from the rentbacks.

Another potential use of public funds worth further investigation is investment in different technologies for living on the coast. One characterization of the problem is that the current housing stock is too durable for the rapidly changing physical environment, and less durable, movable, and modular housing might better match the

environment[53]. However, current incentives do not appear to be transitioning the coastal housing stock in these ways. Evidence for building back bigger suggests current markets are reinforcing excess durability in the coastal zone[17]. Because larger structures tend to be more valuable, building back bigger triggers more investments in defending the shoreline in its current state rather than adapting to a very different future[54]. As such, public investments that subsidize shoreline stabilization may be better spent on incentives for adaptive housing technology.

Although we incorporate a wide range of modeling frameworks and empirical studies, many issues warrant future investigation. C-HOM only includes income as an indicator of coastal gentrification, and future studies might also consider race. For beach nourishment, modeling could allow costs to rise with cumulative sand use[55] or examine the impacts of changing the oceanfront-to-non-oceanfront tax ratio[42]. C-HOM could also be customized to examine other policies and coastal management interventions, e.g. flood insurance, dune building, sea walls, and zoning. The extent to which restrictions placed on institutional investors for short-term rentals (e.g., AirBnB and VRBO) affect outcomes is another important research and policy question. Generalizing the physical system to incorporate shoreline orientation, wave climate, coastline management in neighboring communities, and dune dynamics can all lead to nonlinear erosion and accretion[32–35,56,57], which could induce spatial and temporal spillovers in management decisions. Modeling these elements requires treating space explicitly, in the alongshore and vertical elevation dimensions. Explicit spatial modeling could characterize variation in elevations within the community that leads to different flood risks, model feedbacks across management strategies in neighboring communities that alter shoreline gradients[32,34], and introduce variation in implicit subsidies and uptake of flood insurance. Allowing for climate belief heterogeneity[12,58] in a spatial model could aggregate climate change believers and climate skeptics in separate locations, which through different management strategies could induce complex dynamics in shoreline position and local property markets along the path toward inundation. Lastly, although the barrier island communities that motivate our modeling are largely built out and have been for decades, allowing the housing supply to adjust upward or downward would be a useful generalization for future work. Endogenizing housing supply would enable C-HOM to analyze a wider range of coastal communities.

## Methods
### C-HOM overview
C-HOM combines elements from theoretical, numerical modeling, and empirical literature in economics, finance, nonlinear systems, and coastal processes.

First, we adopt an asset-price approach that links the sales price of a property to the capitalized flow of housing market rent[46,59]. We specifically modify the user cost of housing model to parameterize variation in incentives as a function of demographics, the changing physical environment, and broader economic conditions[46,60]. By linking housing purchases to the flow of coastal amenities and risks, the user-cost approach introduces dynamics that result from different incentives faced by owners, institutional investors, and renters. These differences could become more important as climate change alters the physical environment.

Second, we model feedbacks in the coupled human-natural coastal system[29,33,35,41,61]. The natural system affects the human system, which in turn affects the natural system[62]. Specifically, the physical environment changes with erosion, SLR, and storms. These changes affect risks and values of coastal real estate. Human responses, including beach nourishment, create feedbacks between the human and natural systems[33,42] (Fig. 2).

Third, we use agent-based modeling to capture nonlinearities and heterogeneity in the coupled human-natural system[58,63]. In contrast to homogeneous agents with complete information, our model agents have heterogeneous perceptions of and tolerance for environmental risk, and agents use finite-time forecasts of future returns and coastline position to make investment and policy decisions. Nonlinearity is incorporated via positive feedbacks in property value appreciation such that agents can push prices away from equilibrium in the short run, which can lead to transient bubbles in real estate markets[64]. A bubble occurs when prices rise substantially above expected prices based on market fundamentals over a short period of time. In the long run, arbitrage opportunities move the property market back towards equilibrium.

Fourth, we incorporate flows of amenity values and climate risks based on empirical valuation studies in environmental economics[55]. Coastal housing markets respond to changes in environmental conditions, and equilibrium market outcomes reveal peoples' preferences and perceptions of coastal amenities and risks, as well as expectations of climate impacts. Empirical estimates from coastal housing markets show that proximity to the shoreline and coastal wetlands[65], beach width[39], and beach views[66] increase property values. Exposure to storm risk and flood hazards tend to decrease home values[67].

Fifth, C-HOM reflects findings that public investments in adaptation and risk mitigation capitalize into real estate prices. Risk reductions from a range of climate adaptation infrastructure are reflected in nearshore housing prices[68], including beach nourishment[69], construction of vegetated dunes[70], and hard structures, such as sea walls or revetments[71]. Even sandbags can temporarily protect properties and stabilize shorelines. Because investments in coastal adaptation are often federally subsidized in the US, local housing market prices are less volatile relative to the increase in risk[12]. Federal subsidies for beach nourishment can even create a bubble in coastal real estate markets such that removing subsidies would substantially decrease coastal property values[29].

The base model, to which other features are added, is an asset-price model of investment in housing that depends on the flow of housing services (rent) and the opportunity cost of ownership (user cost). The user cost includes cost of capital (the discount rate), depreciation, a risk premium for the real estate market, expected capital gains, and interactions with income and property tax rates that also affect the capitalization of rents[46,60]. We generalize the model to incorporate the economic value of environmental amenities and risks associated with climate change. In contrast to the standard user cost model, we account for heterogeneity in risk preferences within and across resident owners, resident renters, and an outside institutional investor. This accounts for the ways that peoples' desire to live at the coast and appreciation of environmental amenities intersect with risk of inundation from storms and other elements of the user cost model.

Capital asset models of housing provide the conceptual relationship between stock and flow variables that determine asset price, namely property sales price ($P$) and property rent ($R$). Sale price represents the stock value whereas rents capture the flow value of housing services. The value of an asset is determined from the (expected) flow value, or rents, accrued over time such that the present value of an asset is equal to the discounted sum of future rents. Over an infinite horizon and with constant $R$ and capitalization rate, $i$, the stock value reduces to:

$$P = \sum_{t=1}^{\infty} \frac{R_t}{(1+i)^t} = \frac{R}{i} \tag{1}$$

Because discounting weighs terms according to exponential decay, the infinite-horizon problem approximates the value of a durable asset, such as real estate, that is expected to be long-lived.

Equation (1) forms the conceptual basis for the user cost of housing model[46]. The user-cost equivalence is a capital-theoretic relationship equating the marginal cost of owning capital to the rate of

return. The cost to own one unit of housing for one year is a percentage $i$ of the total present value $P$, and the return on capital investment is the rent $R$ paid to consume those housing services over the same period. Owner occupants similarly receive a flow of housing services valued at $R$. In order to simulate coastal housing markets confronted with climate change, C-HOM decomposes the numerator and denominator in Eq. (1) and endogenizes a subset of the individual pieces.

## User cost of housing

In the user cost of housing model, the capitalization rate is:

$$i = \delta(1 - \tau_t^{inc}) + \tau^p(1 - \tau^{inc}) + \gamma + r^p - E[g_{t+1}] \qquad (2)$$

The discount rate, $\delta$, represents the interest rate on a mortgage. In the US market, this rate is modified by the marginal income tax rate $\tau^{inc}$ because homeowners can deduct mortgage interest, which effectively decreases the discount rate; note that this will depend on an individual's marginal tax rate. Property taxes are also deductible such that individuals are not double taxed on income used to pay local property taxes, and this too is proportional to the marginal tax rate for an individual's income bracket, $(1 - \tau^{inc})$. Capitalization also includes depreciation on the physical structure, $\gamma$, which in our model we assume is constant and captures maintenance and repair costs (assumed separate from climate risks). The risk premium, $r^p$, reflects the opportunity of investing in a risky asset (housing in our case), and we further parameterize this as a function of the physical system. Expectations of future capital gains are in $E[g]$. If one expects house prices to increase by X% over the course of holding it for one period, then the capital gain effectively decreases the cost of owning the asset over that period, and vice versa for an expected decrease in property value. Combining equations 1 and 2, the user cost model is:

$$P = \frac{R}{(\delta + \tau^p)(1 - \tau^{inc}) + \gamma + r^p - E[g]} \qquad (3)$$

Equation (3) is the basis for C-HOM. The model consists of a fixed number of properties (i.e., the supply of housing is fixed), a pool of agents who generate bid prices for one unit of housing based on Eq. (3), and an investor agent who can purchase multiple units of housing based on the current schedule of agent bid prices and an investor user cost equation similar to Eq. (3).

Our model community is composed of a fixed population of economic agents representing individual households. We only consider residential properties in the model. Because the number of properties and the number of agents who could own or rent those properties is fixed, when a new agent enters the market, the agent necessarily triggers the exit of another agent. For example, when higher income agents enter, lower income agents exit by assumption. We do not model where the exiting agents go and assume that they are absorbed by real estate markets outside the modeled coastal community.

During each model time step, every agent chooses to either purchase one unit of housing in the market or rent one unit of housing from the investor agent. The investor can potentially own all, some, or none of the housing units each year. One can think of the investor agent as a bank or a pension fund that owns coastal real estate as part of its investment portfolio, and it pays a fee to a management company to rent out the properties. For simplicity, there is one investor agent representing a vast pool of potential institutional and individual investors external to the coastal community being modeled. Every agent formulates a unique price and rental bid for housing based on the user cost of housing and value of consumption flow described in more detail below. Some components of the user cost formulation are unique to each agent (e.g., the income tax bracket), while other components are common (e.g., the housing depreciation rate).

The primary model variables describing the state of the system are the equilibrium price and investor market share. These are established by a market clearing condition described below. Lastly, communities manage the beach, to enhance its recreational value and provide some protection from storms, using a combination of government subsidies and self-financing. To self-finance, the resident owners vote on when and how often to nourish. Lastly, a coastal community is geographically divided into two market segments. The oceanfront market is defined as the single row of houses directly adjoining the beach, and the non-oceanfront market is the remainder of the coastal community. Non-oceanfront properties benefit from wider beaches but not as much as oceanfront properties. We model the supply of housing as fixed because most coastal communities along the East and Gulf Coasts are built out and, when storms destroy structures, typically structures are built back. This also means that we model the effects of storm risk on property values and not the episodic nature of actual storm realizations and resulting damages.

The model spatial resolution is smaller than a town but larger than a census block group; we refer to this as a nourishment unit. This spatial extent allows for the decision-making unit for a nourishment project that is funded by property taxes to be a town, but it includes the possibilities that the town may not nourish its entire beach and that there are special tax districts within the town that pay higher rates to fund the project[42]. The differences between oceanfront and non-oceanfront markets are embedded in fixed parameters discussed below (e.g., there is a base risk premium, which is higher for front row agents) and in how beach management is financed, including a political economy in which the front row pays a higher share of costs[42]. Agents choose when to nourish the beach width. Extending the beach increases the agent's rent by increasing their willingness to pay.

## Agent-based user cost

We adapted the user cost of housing model for a household in Eq. (3) to simulate a coastal property market comprised of a population of agents seeking to reside at the coast as either a resident renter or resident owner. There is also one non-resident property investor agent (referred to hereafter as the investor agent, with superscript $I$) that may purchase many units of housing. The potential owner agents (hereafter referred to as owner agents, with superscript $O$) are indifferent to owning versus renting beyond incentives captured by the model, and we assume that owner agents who do not end up owning instead rent from the investor agent. Thus, the owner agent enters the market willing to either rent or own.

There are $n$ potential owner agents and $n$ properties. The user cost of housing model determines each agent's rent and corresponding price bid. Owner agents are indexed with the subscript $j$. The investor agent considers all potential owner rent and price bids to determine a rent offer based on the user cost of the investor. The investor rent offer is compared to the list of owner agent rent bids to determine what fraction (share) of housing units can be purchased with an owner agent willing to rent at the rate offered by the investor. This fraction is referred to as the investor market share. This calculation determines the equilibrium house price and the share of the n available properties an investor will own.

Owner agent $j$'s rental bid is composed of their willingness pay for coastal amenities, $WTP_j$, and the annualized value of housing services, $HV$:

$$R_{j,t}^{O,bid} = WTP_{j,t} + HV \qquad (4)$$

$HV$ is a constant (same for all agents) and $WTP_{j,t}$ is drawn randomly and updated over time according to demographics of the agents. We abstract away from short-term vacation rentals and assume that these

values are captured by the flow value of *WTP* in the prospective owner rent bids. As described below, $WTP_{j,t}$ is further decomposed to incorporate the influence of beach width on each agent's willingness to pay to live at the beach. Importantly, *WTP* is an additional amount beyond the fixed value of housing services. It reflects preferences for coastal living in general as well as preferences for specific coastal amenities such as the width of the beach, and it is correlated with income. Rewriting the user cost Eq. (3) to include indices for parameters that vary between agents, and replacing the rent *R* with the decomposed rent Eq. (4) gives the owner agent price bid function:

$$P_{j,t}^{O,bid} = \frac{WTP_{j,t} + HV}{(\delta + \tau_t^p)(1 - \tau_t^{inc}) + \gamma + r_{j,t}^p - E_{j,t}^O g_{t+1}} \quad (5)$$

The term $\tau_j^{inc}$ is unique to each agent and drawn randomly, where the parameters of the distribution evolve over time based on how the local market changes relative to other housing markets (details are described below). An agent's $WTP_j$ and $\tau_j^{inc}$ are correlated such that agents in a higher tax bracket (larger $\tau_j^{inc}$) also have a larger $WTP_j$ to reflect the influence of income on *WTP*. The correlation, however, is assumed to be imperfect to allow heterogeneous preferences for coastal living to influence housing markets and outcomes independent of the income channel.

Next, owner agent bid prices are sorted from lowest to highest and corresponding rent bids are sorted according to the ordered list of bid prices (Supplementary Fig. S5). The sorted rents are not monotonically increasing like the bid prices because $\tau_j^{inc}$ has a stochastic component. Nevertheless, the sorted rents trend upward as a result of the influence of income on *WTP*.

The resulting investor market share is the highest share of the market for which the investor can outbid potential owners to purchase properties and still rent the properties to prospective owners with zero vacancy. This effectively requires that investors offer a lower rent than the rent bid associated with a prospective owner's bid price. If the investor purchases 100% of the market, this entails the investor outbidding the highest bid price of prospective owners, while the investor's associated rent bid is also below all owner rent bids. If this condition does not hold, the investor iterates through each market share until the condition does hold. That is, the investor finds a market share for which it outbids the highest owner bid price and the investor's rent bid is below all rent bids for non-owners, which is needed to ensure zero vacancy.

The investor's rent offer is based on the user cost equation in 5 with key differences that the investor has a corporate tax rate $\tau^{I,c}$ in place of $\tau_j^{inc}$ and an additional term *m* subtracted from the rent to reflect the cost of property management. Rearranging terms, rent offers are:

$$R^{I,offer} = P_{j,t}^{O,bid}[(\delta + \tau_t^p)(1 - \tau^{I,inc}) + \gamma + r_{I,t}^p - E_t^I g_{t+1}] + m \quad (6)$$

For each property, the rent offered by the investor is then compared against the prospective owner agent's rent bid. If

$$R^{I,offer} < R_j^{O,bid} \quad (7)$$

then the investor codes this property as occupied and decides to buy it. This process is iterated from $j = 1$ to $n$. When the investor rent offer compared against agent *j*'s rent bid no longer satisfies the inequality in (7), the property is coded as vacant, and it will be purchased and occupied by an owner resident.

## Beach and elevation dynamics
We abstract from short time-scale processes that unfold daily or seasonally and use a linear erosion term ($\psi_{bw}$) to represent the long-term erosion signal. Depending on the outcome of beach nourishment

decisions (see Shoreline Management below), additional beach width ($\Delta_t$) can be added. The state equation describing the width of the beach ($bw$) is then:

$$bw_{t+1} = bw_t - \psi_{bw} + \Delta_t \quad (8)$$

The state equation for mean seal level (*MSL*) also assumes a constant linear rate of increase ($\psi_{SL}$):

$$MSL_{t+1} = MSL_t + \psi_{SL} \quad (9)$$

## Willingness to pay
Beach width influences an agent's willingness to pay to live at the beach[39,72]. Therefore, $WTP_{j,t}(4)$ is composed of a base willingness to pay and a beach width-dependent willingness to pay. The base willingness to pay component $WTP_j^{base}$ reflects coastal amenities that are not directly tied to the beach width. The width-dependent component consists of the average expected future beach width $E[bw_{t+1}]$, a hedonic parameter ($\beta$), which reflects the marginal value of the beach width attribute, and a base parameter $\alpha_j$ scaling the contribution of beach width to the monetized total willingness to pay, which theoretically can be estimated in a first-stage hedonic model[39,73]. Both $\alpha_j$ and $WTP_j^{base}$ are randomly distributed across agents, again with parameters of the distribution changing over time based on demographics of the agents. The value of $\beta$ is larger for agents in the oceanfront market such that there are separate parameters, $\beta^{OF}$ and $\beta^{NOF}$.

$$WTP_{j,t} = WTP_j^{base} + \alpha_j E[bw_{t+1}]^{\beta} \quad (10)$$

The expected beach width term is calculated by averaging the beach width over the previous $t_{bw}$ years:

$$E[bw_{t+1}] = \frac{1}{t_{bw}} \sum_{n=1}^{t_{bw}} bw_{t-n} \quad (11)$$

## Risk premium
All agents are subject to risks from SLR and storms. These risks are included in a risk premium term that includes background risk associated with investment in real estate markets and a dynamic agent-specific term that reflects changing physical risks, heterogeneous climate beliefs, and risk tolerance. In the user cost framework, the risk term effectively augments the discount rate. In our context, this implies that property values are a smaller multiple of rents when physical risks are larger or risk tolerance is lower. Intuitively, this is the same mechanism that exerts downward pressure on property value when interest rates rise.

The risk premium term for agent *j* at time *t* is:

$$r_{j,t}^p = \overline{r^p} + [r^{OF} + r_t^{St} + r_t^{SL}]\pi_{j,t} \quad (12)$$

where $\overline{r^p}$ is background risk in real estate markets, and the three terms inside the square brackets capture climate risks. Specifically, $r^{OF}$ captures the reality that oceanfront properties are more exposed to risk than properties behind them, $r_t^{St}$ captures risk from storms, $r_t^{SL}$ captures the risks from SLR that include inundation risk, how SLR increases risks from storm surge, and the effects of SLR on sunny day flooding. The sum of the three physical risk terms in the square brackets is then modified by $\pi_{j,t}$, a risk multiplier that combines individual beliefs and tolerance. If an agent is risk neutral and has beliefs that match the objective risks, $\pi_j = 1$ such that $r_{j,t}^p$ includes actuarially fair climate risks, and we assume $\pi_j = 1$ for the outside institutional investor[13]. We assume that these parameters also reflect the cost of insurance, which

capitalizes into property price, and this cost does not vary within the model community.

We assume that $r^{OF}$ is fixed through time, which implies that there is a permanently higher risk of oceanfront property but that both oceanfront and non-oceanfront properties experience the other physical risks similarly. We parameterize $r_t^{St}$ to reflect the episodic nature of storms, $r_t^{St} = \frac{a_1}{\lambda_t}$, where $\lambda_t$ is the average storm return interval over the past 30 years, and $a_1$ is a scaling parameter. To account for risk of SLR, we parameterize $r_t^{SL}$ to be a function of property elevation relative to sea level. Specifically,

$$r_t^{SL} = a_2(1 - (h^{elev} - MSL_t))^n \tag{13}$$

where $MSL_t$ is mean sea level, which changes with SLR, $h^{elev}$ is initial barrier elevation, $a_2$ is a scaling parameter, and $n$ controls the non-linearity of risk as a function of $MSL_t$.

## Agent distribution adjustments and outside markets

An important driver in real estate markets is demographic change, and because our model represents coastal real estate markets over long time horizons, we account for demographic changes by updating agent parameter distributions. To this end, we use interactions with outside markets to benchmark these changes. For both oceanfront and non-oceanfront properties, there is a corresponding external market value for housing that serves as a benchmark for owning coastal property in the community, given by $P_e^{OF}$ and $P_e^{NOF}$. These external market prices are exogenous and can be thought of as reflecting large-scale real estate market trends in other communities with appropriate amenity adjustments. That is, $P_e^{NOF}$ is a representative inland property price with sufficient amenities to compensate for the differential between coastal and non-coastal average property values, and $P_e^{OF}$ is an inland property price with sufficient amenities to compensate for the oceanfront and coastal premia. Examples of amenity adjustments to make inland properties equivalent include being in a particularly good school district, on a lakefront, near green space, or in close proximity to cultural amenities. Deviations in coastal property values from $P_e^{OF}$ and $P_e^{NOF}$ drive changes in the owner agent population by shifting the mean income and coastal willingness to pay distributions up or down as property values attempt to equilibrate with the outside markets. Shifts in the distribution of agents lead to a flux of new agents with higher or lower income and possibly different beliefs into the coastal market (and a corresponding flux of agents out) in response to the value of coastal real-estate relative to the outside market.

There are two possible feedbacks in the coastal market due to the external market, one based on arbitrage and the other based on herding. Specifically, over long time scales, there is a sluggish negative feedback component driving the average income distribution and coastal willingness to pay distributions of owner agents up or down, with the effect of driving coastal property value toward $P_e$ (price arbitrage). Over short time scales, there is a faster and positive feedback that can drive income and *WTP* distributions in the opposite direction with the effect of temporarily driving property values away from $P_e$ (herding behavior). The specific analytical forms are based on Dieci and Westerhoff[64].

Let $A_t$ be a parameter of a Beta distribution that changes according to arbitrage opportunities with property values in outside markets. We fix one shape parameter and allow the other parameter to adjust based on these arbitrage opportunities and control the concentration of probability mass along the [0,1] interval. For each of $n$ agents in each period $t$, we draw a $(4 \times 1)$ vector from the Beta distribution. We then convert these draws on the unit interval to draws from our parameters of interest by

re-scaling them based on upper and lower bounds:

$$\tau_{j,t}^{inc} \in [\tau^{inc,L}, \tau^{inc,U}] \tag{14}$$

$$WTP_{j,t}^{base} \in [WTP^{base,L}, WTP_t^{base,U}] \tag{15}$$

$$\alpha_{j,t} \in [\alpha^L, \alpha_t^U] \tag{16}$$

$$\pi_{j,t} \in [\pi^L, \pi^U] \tag{17}$$

The upper and lower limits of the income tax parameters do not change over time. They are fixed based on the current US federal tax code. We also fix the bounds of the risk tolerance parameter $\pi$. We fix the lower bounds of $WTP^{base}$ and the beach width scaling parameter $\alpha$ but allow their upper limits to increase or decrease based on the adjustment process described below.

The agent adjustment equation for $A_t$ that can move the distribution in either direction depending on the sign of $P_t - P_t^e$:

$$A_t = A_{t-1} + \phi(W_t(P_t - P_t^e) + (1 - W_t)(P_t^e - P_t)) \tag{18}$$

The term $W_t$ is the strength of the short-term positive feedback associated with herding, and $(1 - W_t)$ is the strength of the longer more sluggish negative feedback that drives prices back to equilibrium:

$$W_t = \frac{1}{1 + h(P_t - P_t^e)^2} \tag{19}$$

The parameter $h$ gives the speed of switching between the two types of feedback and the parameter $\phi$ controls the how quickly the agent distribution changes, and hence how quickly agents can flux in and out of the market. For a given deviation between coastal and non-coastal property values, a larger $h$ value initiates the switching from positive to negative feedbacks more quickly. If $h$ is very high, $W_t$ approaches zero, and the arbitrage effect dominates. Effectively, this means that, ceteris paribus, if coastal properties are undervalued relative to the external market, higher-income agents enter and prices increase, whereas if coastal prices are overvalued, higher-income agents exit and prices drop. Similarly, if coastal properties are undervalued relative to the external market, agents enter who have higher *WTP* for coastal living, higher *WTP* for beach width, and higher risk tolerance (to reflect higher-income agents with less relative wealth at risk).

The upper limits of $WTP^{base}$ and $\alpha$ adjust based on the percent difference in property prices within the community compared to the outside market:

$$WTP_{t+1}^{base,U} = WTP_t^{base,U}\left(1 + \frac{(P_t^e - P_t)}{P_t^e}\right) \tag{20}$$

This allows for the possibility that an agent could have a very high *WTP* for the experience of coastal living (in the numerator of the user cost equation) despite growing risks from SLR (in the denominator of the user cost equation). Similarly, the upper limit for the scale parameter of the value of beach width adjusts based on outside markets:

$$\alpha_{t+1}^U = \alpha_t^U\left(1 + \frac{(P_t^e - P_t)}{P_t^e}\right) \tag{21}$$

## Expected capital gains

Agents form heterogeneous expectations of capital gains based on past values of market returns[74]. Specifically, each agent evaluates a price return over time, where the time of measuring the return varies

between one and 30 years. Agents are randomly assigned one of the thirty price/return time scales to form their expected capital gains. The expected capital gains at the current moment for a given agent is simply the return for one year that would yield the total return the agent found over their assigned time scale. For example, an agent assigned 30 years calculates the annualized capital gain or loss at $t$ based on the ratio of realized property prices 30 years before the previous period ($P_t - 31$) to the previous period ($P_t - 1$). In this way, some agents are reactionary as they adjust expected returns over short time scales, while others are more sluggish in computing returns over a long time scale. We assume that foresight about future climate change is captured in the risk parameters, and as a practical matter, the influence on owner bids cannot be separately identified from in the denominator of equation (5). We calculate the investor's expected capital gain in each period as the median of owner expected capital gains.

## Shoreline management

Previous models of beach nourishment decisions show that the human and natural systems jointly determine shoreline position and coastal property values[17,31,33,75,76]. When a community decides to nourish, they commit to a nourishment plan defined by periodic nourishments over a ten-year horizon. A nourishment plan is chosen from a schedule of possible intervals. Nourishment is funded locally through municipal bonds that are paid for by the residents over the first 5 years of the plan through a temporary (five-year) increase in property taxes ($\tau^p$). This nourishment decision framework is similar to the one used in Duck, NC[77] and described in Mullin, Smith, and McNamara[42].

A ten-year nourishment plan that will subsequently be voted on is chosen by comparing the property tax burden (cost) against the increase in property value (benefit). A given nourishment plan consists of a series of scheduled nourishments occurring over a ten-year period and an increase in property tax lasting five years. If the benefits minus the costs are negative, then no nourishment plan is chosen - in this case the nourishment plans are re-evaluated every time step until a nourishment plan is chosen. Overlapping nourishment plans and property tax adjustments are permitted so long as they do not conflict with previous plans by scheduling two nourishment events for the same year or result in consecutively scheduled nourishments (i.e., nourishing every year).

Plan costs and benefits are tabulated each step for a range of nourishment intervals (ranging from every other year to every five years) leading to a menu of nourishment options on which agents vote. For each proposed plan option, an average expected beach width over a 30-year time horizon is calculated based on the estimated shoreline retreat rate (from the previous 30 years), and the quantity of sand needed for each nourishment event is tracked with the cost of anticipated nourishment costs occurring later in the plan discounted at a rate $\delta$.

The cost of a nourishment event at some time $t$ is determined from

$$cost(t) = f + c(bw_o - bw_t)) * L * D \qquad (22)$$

where $f$ is the fixed cost of nourishment, $c$ is the cost of sand per $m^3$, $bw_o$ is the nourished beach width, $L$ is the alongshore length of the nourishment, and $D$ is the depth of the shoreface.

The total cost ($TC$) of option $i$ (where $i$ is the nourishment interval) is determined as the sum of the discounted fixed and variable costs of nourishment:

$$TC_i = \sum_{t=1}^{10} \frac{cost_i(t)}{(1+\delta)^{t-1}} \qquad (23)$$

A 5-year amortization schedule is then used to determine the total yearly cost of the loan repayment ($TCY_i$):

$$TCY_i = TC_i \frac{\delta(1+\delta)^5}{(1+\delta)^5 - 1} \qquad (24)$$

Oceanfront homes take on a greater share of the nourishment costs. The additional property tax rate is determined by setting the total cost of nourishment per year to the sum of the taxes collected amongst all agents in both the oceanfront and non-oceanfront markets:

$$TCY_i = \sum_{j=1}^{n} \rho \tau^{p,add} I_j^{OF} P_{t-1}^{OF} + \tau^{p,add}(1 - I_j^{OF}) P_{t-1}^{NOF} \qquad (25)$$

where $\rho$ is the tax ratio of oceanfront to non-oceanfront, the indicator variable $I_j^{OF} = 1$ for oceanfront homes and 0 for non-oceanfront homes, and $\tau^{p,add}$ is the additional property tax increment. For example, if a nourishment plan is chosen, then the property tax for oceanfront homes is $\tau^p + \rho\tau^{p,add}$, and for non-oceanfront homes is $\tau^p + \tau^{p,add}$.

The total benefit of each interval/option is estimated by forward simulating property value via the user cost model. The forward simulation accounts for the average beach width under the proposed nourishment scenario that enters the coastal willingness to pay term, and the property tax, including the nourishment tax adjustment (and adjustments for previous nourishments if still applicable), enters into the denominator of the user cost model. The benefit is the increase in property value compared to the case of no nourishment, which will have both a lower expected beach width and lower property tax rate. Each agent evaluates whether the property value increase is greater than the extra tax burden. If yes, then the agent votes to nourish. The proposed nourishment plan is implemented if at least 50% of resident-owner agents vote to nourish. Importantly, institutional investors do not vote but are still taxed if nourishment is approved.

Nourishment projects typically involve dune building in conjunction with widening of the beach. We do not explicitly model dune building and assume that benefits and costs of dune construction are captured implicitly in the nourishment decisions. We leave explicit modeling of dune dynamics as a future extension of the model.

Supplementary Table 1 summarizes model parameters, provides details on how they were chosen, and motivates the sensitivity analysis to explore alternative parameter values, different functional forms, and different scenarios. The model was coded and run in Matlab R2022b.

## Reporting summary

Further information on research design is available in the Nature Portfolio Reporting Summary linked to this article.

## Data availability

Simulation data for Figs. 3, 4, 5, and 6 that support the findings of this study have been deposited in https://github.com/dylanmcn/C_HOM[78].

## Code availability

Numerical model code can be accessed at https://github.com/dylanmcn/C_HOM[78].

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

## Acknowledgements

This research was supported by the National Science Foundation (NSF) under Grant no. CNH 1715638: D.E.M., M.D.S., S.G. and C.E.L.

## Author contributions

D.E.M. developed the model, designed the numerical experiments, analyzed the results, edited the code, and wrote the paper. MDS developed the model, designed the numerical experiments, analyzed the results, and wrote the paper. Z.W. developed the model, wrote the code, and wrote the paper. S.G. developed the model, conducted the analysis in Fig. 1, and edited the paper. C.E.L. developed the model and edited the paper.

## Competing interests

The authors declare no competing interests.
