## [Peer Review File · Nature Communications]

Policy and Market Forces Delay Real Estate Price Declines on the US CoastVIEWER COMMENTS

Reviewer #1 (Remarks to the Author):

This paper develops the Coastal Home Ownership Model (C-HOM) to evaluate the long-term effects of sea level rise (SLR) on coastal real estate markets. C-HOM integrates several important features of real estate markets, physical coastal systems, and their interactions to explain how changes in the real estate markets and SLR mitigation strategies (in this case, beach nourishment) might support (or delay the eventual collapse) of coastal housing markets. The method combines evidence/methods from several strands of the existing literature to model the human-physical integrated system.

This is a valuable exercise demonstrating how real estate markets may not reflect SLR risk due to endogenous changes in investment behavior and demographic characteristics. My main substantive comment relates to the feature of the model that housing supply is fixed and the demand is the driver of prices. In the long run, it seems that supply is another reasonable margin of adjustment. The authors justify that coastal communities are often built back after structures are destroyed, but might this decision be endogenous to impending SLR and the changing environment as well? If it is out of scope to build this into the current framework, then perhaps some discussion of its impacts would be useful.

A second comment relates to the values used for model inputs. For some model parameters, the range of appropriate values may be wider, e.g., the discount rate. Is there anything interesting that might happen with the predictions when varying these values or allowing for heterogeneity here?

Pg. 3: "Empirical data are also necessarily limited by experiences in the past and may not reflect future scenarios." I am not entirely clear what this means. Do you mean that the individual behaviors generating data might be backward looking (e.g., as in your assumptions of expectations for beach width and capital gains) rather than forward looking?

There are some typographical errors throughout the manuscript that should be corrected.

Reviewer #2 (Remarks to the Author)

Review of Manuscript NCOMMS-23-32586

Policy and Market Forces Delay Inevitable Real Estate Price Declines on the Coast

This paper introduces the Coastal Home Ownership Model (C-HOM) and conducts an in-depth analysis of the long-term evolution of coastal real estate markets.

Key Points for Consideration:

1. **Structural Order:** The paper's structural arrangement could benefit from reordering. It would be more logical and enhance readability if the Model section is repositioned to precede the Results and Conclusion sections.
2. **Parameter Validation:** An essential aspect of assessing the validity of the C-HOM model is comparing the parameter values presented in Table 1 with real-world data. This validation process helps ensure that the model accurately mirrors real-world scenarios.
3. **Local Market Dynamics:** Recognizing the localized nature of real estate markets, it is crucial to address how the C-HOM model accounts for regional variations. Incorporating these local differences is essential for the model's applicability across diverse markets.
4. **Confidence Interval Inclusion:** To enhance the informativeness of the results, it is recommended that the C-HOM model incorporates confidence intervals. This addition would provide a more comprehensive understanding of the statistical significance of the findings, thereby enhancing the robustness of the analysis.

These suggestions aim to enhance the paper's overall clarity, validity, and applicability, contributing to a more comprehensive exploration of coastal real estate market dynamics.

Reviewer #3 (Remarks to the Author):

This manuscript presents the Coastal Home Ownership Model, which uses an agent-based modeling approach to simulate the feedbacks in the coupled human-natural system in coastal communities experiencing emerging flood exposure due to sea level rise. The model is applied to examine how owners, renters, and investors value coastal property and invest in coastal management under various policy scenarios. Results indicate that certain management policies, such as providing subsidies for beach nourishment and tax advantages for high-income owners, will dampen and delay the effect of sea level rise on property values in coastal communities. Removing subsidies will allow property values to more accurately reflect the risk due to sea level rise but will also lead to a transition to wealthier ownership along the coast, pushing out lower-income owners. This highlights an interesting trade-off in the management of coastal property markets, with important economic and equity implications in how coastal communities respond to sea level rise. Overall, the manuscript is well-written, clearly organized, and addresses a topic of current interest to academics and practitioners across multiple fields. There are a few points of clarification that should be addressed before recommending the article for publication.

Line numbers were not provided, so I will do my best to be clear about the relevant locations in the text.

Major comments:

- The introduction should include some review of the literature on coupled human-natural systems modeling and clearly distinguish what advances are achieved through C-HOM. This would enable the reader to better understand the novelty of the work.
- The manuscript presents three scenarios that explore how changes from the baseline will influence the feedbacks observed in the system (page 6, paragraph 1). However, no rationale was provided to support the choice of the values for the beach nourishment subsidy and the timing/magnitude of outside market appreciation/depreciation. I think the paper would benefit from a sensitivity analysis that explores how changes in the choice of these values influences the system trajectory.
- Certain modeling components should be explained in greater detail to enable the reader to evaluate the approach and understand potential limitations. For example:
 - o How is the volume of sand (and thus the price of nourishment) determined over time?
 - o Does beach nourishment have any effect on the level of exposure (i.e., can a wider beach reduce some flooding impact)? Or is beach nourishment only viewed from an amenity perspective? If the latter is true, which I believe is the case, is it possible to incorporate hazard reduction due to nourishment in the risk premium formulation?
 - o A nourished dune height is listed in Table S1, but I don't recall this being defined in the model. Please clarify if dune height is considered in the model.
 - o Please provide an explanation of how the scaling parameters (a_1 and a_2) were determined.

Minor comments:

- Page 4, paragraph 1: Please provide a definition for a "bubble" in the context of this paper. While this term is widely used, I think a clear explanation would be helpful for the general reader.
- Page 4, paragraph 5: Why is a 150-year time horizon chosen for this analysis?
- Page 6, paragraph 2: Is the "barrier height" a reference to the barrier island itself? Or is there assumed to be another structural barrier providing flood protection? Please clarify.
- Page 17, paragraph 1: How is the spatial extent of the nourishment unit determined? For this hypothetical case, why is it important that it be "smaller than a town but larger than a census block group"?
- Page 18, first line: Include "equation" before the number 4.
- Page 20, equation 13: Please define all terms that are not previously defined. I think a_3 should be a_2 , as referenced in Table S1.

Reviewer #1 (Remarks to the Author):

This paper develops the Coastal Home Ownership Model (C-HOM) to evaluate the long-term effects of sea level rise (SLR) on coastal real estate markets. C-HOM integrates several important features of real estate markets, physical coastal systems, and their interactions to explain how changes in the real estate markets and SLR mitigation strategies (in this case, beach nourishment) might support (or delay the eventual collapse) of coastal housing markets. The method combines evidence/methods from several strands of the existing literature to model the human-physical integrated system.

We thank the reviewer for providing constructive comments. Below we repeat the reviewer's comments in *italics* with our responses in plain text. We believe that addressing these comments has strengthened the clarity and the contribution of the paper.

This is a valuable exercise demonstrating how real estate markets may not reflect SLR risk due to endogenous changes in investment behavior and demographic characteristics. My main substantive comment relates to the feature of the model that housing supply is fixed and the demand is the driver of prices. In the long run, it seems that supply is another reasonable margin of adjustment. The authors justify that coastal communities are often built back after structures are destroyed, but might this decision be endogenous to impending SLR and the changing environment as well? If it is out of scope to build this into the current framework, then perhaps some discussion of its impacts would be useful.

The reviewer is correct that the model fixes the housing supply, and it is beyond our scope to endogenize the housing supply. This assumption is justified by our focus on barrier island communities in the Southeastern U.S., which are built out (and have been for some time). What happens when inundation forces the housing supply to shrink substantially is speculative, and our model is primarily used to probe the period leading up to that eventuality. We now acknowledge this limitation in the discussion section where we point to endogenizing housing supply as important future work.

We add the following text to the introduction to clarify:

The barrier island communities in our model constitute a natural boundary to define coastal housing markets and to measure the extent of the impact of local adaptation measures. On developed coastal barrier islands, oceanfront and near shore housing markets are often fully developed and changes in housing supply tend to reflect damage to property and building back homes after hurricanes (Lazarus et al 2018). In a recent empirical study of coastal development on the southern barrier islands in North Carolina, only 8% of parcels were newly developed between 1993 and 2013 (Li, Gopalakrishnan, and Klaiber 2023), with a smaller percentage of developable oceanfront parcels. Holding housing supply fixed in the model allows us to examine demand-driven market dynamics without loss of generality.

Lazarus, E.D., Limber, P.W., Goldstein, E.B., Dodd, R. and Armstrong, S.B., 2018. Building back bigger in hurricane strike zones. *Nature Sustainability*, 1(12), pp.759-762.

Li, Xiaoyu, Gopalakrishnan, S., and Klaiber, H.A., "Local Adaptation and Unintended Coastal Vulnerability: The Effect of Beach Nourishment on Residential Development in North Carolina, Forthcoming, *Journal of the Association of Environmental and Resource Economists*

Then we add the following text to the discussion section:

Although the barrier island communities that motivate our modeling are largely built out and have been for decades, allowing the housing supply to adjust upward or downward would be a useful generalization for future work. Endogenizing housing supply would enable C-HOM to analyze a wider range of coastal communities.

A second comment relates to the values used for model inputs. For some model parameters, the range of appropriate values may be wider, e.g., the discount rate. Is there anything interesting that might happen with the predictions when varying these values or allowing for heterogeneity here?

In response to this comment and the comments of the other reviewers, we conduct a number of sensitivity analyses that are included in the supplementary materials. We also include a new annotated parameter table in which the parameters and parameter ranges are more fully documented and that points to the cases in which we run sensitivity analyses. In general, we find that the qualitative conclusions from the baseline model runs and the three scenario experiments that we run are unchanged. What different parameter values do is to alter the magnitudes of some of the changes and alter the timing of changes, e.g. how long the decline in property value is delayed after the onset of SLR.

For the discount rate example specifically, we used 6%, which is used in the beach nourishment simulations in Gopalakrishnan et al. (2011). The average 30-year nominal mortgage rate from 1971-2023 is 7.74% based on 30-Year Fixed Rate Mortgage Average in the United States, Percent, Weekly, Not Seasonally Adjusted. The average rate of inflation in this time is 4%, suggesting a real mortgage rate close to 3.75%. To explore lower (and potentially higher) rates, we run models using discount rates of 3% and 9%.

We add the following text and figures to the supplemental:

We re-run the nourishment policy experiment using a low discount rate (3%) and a high discount rate (9%). Compared to our findings in the main text, nothing changes qualitatively. We see that the decline in property value begins later for the low discount rate case. This timing difference is due to the fact that, with the lower discount rate, there is more room in the market for high-income owners to flux in (because they are less tax advantaged compared to the cases with higher discount rates).

Low Discount Rate – Reduced Subsidy

High Discount Rate – Reduced Subsidy

To ensure that the mechanism is working as we expect, we also run the low and high discount rate cases relative to the baseline 6% discount case. Here we see clearly that, with the low discount rate, property values are slightly higher before the onset of SLR and more so for non-oceanfront because they are taxed less to fund nourishment. Before SLR, owners are lower-income relative to the baseline because the lower discount rate creates less tax advantage (recall discount rate is multiplied by marginal tax rate). After the onset of SLR, wealthier owners flux in and drive prices up further. The opposite is true in the high discount rate case – lower prices due to less capitalization and more tax advantage for high-income owners so fewer high-income owners who can flux in later.

Low Discount Rate Case – Compared to Baseline

High Discount Rate Case

Pg. 3: "Empirical data are also necessarily limited by experiences in the past and may not reflect future scenarios." I am not entirely clear what this means. Do you mean that the individual behaviors generating data might be backward looking (e.g., as in your assumptions of expectations for beach width and capital gains) rather than forward looking?

We reworded to clarify and replaced that sentence with these two:

The range of experience of SLR and storm risk captured in empirical studies may not include the full range of possibilities under future scenarios. For example, if risks respond nonlinearly as SLR progresses beyond what has been observed in the past, only modeling studies are capable of exploring the implications.

There are some typographical errors throughout the manuscript that should be corrected.

After revising the text, we went through to check two additional times. Thank you again for these comments.

Reviewer #2:

This paper introduces the Coastal Home Ownership Model (C-HOM) and conducts an in-depth analysis of the long-term evolution of coastal real estate markets.

We thank the reviewer for providing constructive comments. Below we repeat the reviewer's comments in *italics* with our responses in plain text. We believe that addressing these comments has strengthened the clarity and the contribution of the paper.

Key Points for Consideration:

1. Structural Order: The paper's structural arrangement could benefit from reordering. It would be more logical and enhance readability if the Model section is repositioned to precede the Results and Conclusion sections.

We appreciate this comment, and we are happy to make this change if the Editor allows it. However, the journal specifications place Methods at the end, and our model description is the Methods section. Specifically, the journal website says: "The main text of an Article should begin with a section headed Introduction of referenced text that expands on the background of the work (some overlap with the abstract is acceptable), followed by sections headed Results, Discussion (if appropriate) and Methods (if appropriate)." Yes, this ordering is not the most common way to structure a paper, but it is not unprecedented in other high-impact journals. For example, PNAS uses this order.

2. Parameter Validation: An essential aspect of assessing the validity of the C-HOM model is comparing the parameter values presented in Table 1 with real-world data. This validation process helps ensure that the model accurately mirrors real-world scenarios.

In response to this comment and the comments of the other reviewers, we include a new annotated parameter table in which the parameter values and parameter ranges are more fully documented and that points to cases in which we run sensitivity analyses. The sensitivity analyses that are included in the supplementary materials and focus on parameter for which we have less ability to pin them down empirically. In general, we find that the qualitative conclusions from the baseline model runs and the three scenario experiments that we run are unchanged. What different parameter values do is to alter the magnitudes of some of the changes and alter the timing of changes, e.g. how long the decline in property value is delayed after the onset of SLR.

While further details are in the Supplemental Material, we highlight two examples here: 1) the discount rate, which is nearly always an important parameter in dynamic economic analyses and 2) the flux parameter, which is difficult to measure and for which we thus consider a wide range.

For the discount rate example specifically, we used 6%, which is used in the beach nourishment simulations in Gopalakrishnan et al. (2011). Alternatively, the average 30-year nominal mortgage rate from 1971-2023 is 7.74% based on the 30-Year Fixed Rate Mortgage Average in the United States, Percent, Weekly, Not Seasonally Adjusted. The average rate of inflation in this time is 4% (based on the U.S. CPI All Urban Consumers), suggesting a real mortgage rate close to 3.75%. To explore lower (and potentially higher) rates, we run models using discount rates of 3% and 9%. To this end, see if reducing the nourishment subsidy has the same effect. Compared to our findings in the initial submission,

nothing changes qualitatively. We see that the decline in property value begins later for the low discount rate case. This is due to the fact that, with the lower discount rate, there is more room in the market for high-income owners to flux in (because they are less tax advantaged compared to the cases with higher discount rates).

Low Discount Rate – Reduced Subsidy

High Discount Rate – Reduced Subsidy

To ensure that the mechanism is working as we expect, we also run the low and high discount rate cases relative to the baseline. So, here you can see clearly that with the low discount rate, property values are slightly higher before the onset of SLR and more so for non-oceanfront because they are taxed less to

fund nourishment. Before SLR, owners are lower-income relative to the baseline because the lower discount rate creates less tax advantage (recall discount rate is multiplied by marginal tax rate). After the onset of SLR, wealthier owners flux in and drive prices up further. The opposite is true in the high discount rate case – lower prices due to less capitalization and more tax advantage for high-income owners so fewer high-income owners who can flux in later.

Low Discount Rate Case – Compared to Baseline

High Discount Rate Case

To examine the sensitivity of the model to the flux parameter, we highlight the speed of adjustment under two flux values. The first (4x baseline) leads to rapid adjustment in which the property value adjusts almost instantaneously to outside markets (within 2 years). The second (0.1x baseline) is a slow adjustment that unfolds over 30 years. These two extremes—unrealistically short or unrealistically long speed of adjustment—provide justification for our baseline flux parameter choice.

3. Local Market Dynamics: Recognizing the localized nature of real estate markets, it is crucial to address how the C-HOM model accounts for regional variations. Incorporating these local differences is essential for the model's applicability across diverse markets.

This is an important point for robustness of our conclusions. We explore this issue in two dimensions: 1) the base parameter in the willingness-to-pay function; and 2) the rate of sea-level-rise. We add the following texts to the supplemental:

The base parameter (alpha) captures the variation in the background strength of the real estate market. A higher value means higher-value homes, typically associated with proximity to stronger labor markets, better schools, lower crime, and other environmental amenities that we are not explicitly modeling. Varying the rate of SLR accounts for the fact that SLR has been documented to be heterogeneous along the coast (Pecuch et al. 2018). We allow the base parameter to increase or decrease and the rate of SLR to increase and re-run the model for all four combinations.

Piecuch, C.G., Huybers, P., Hay, C.C., Kemp, A.C., Little, C.M., Mitrovica, J.X., Ponte, R.M. and Tingley, M.P., 2018. Origin of spatial variation in US East Coast sea-level trends during 1900–2017. *Nature*, 564(7736), pp.400-404.

First, we show how these changes do not affect the qualitative conclusions of the nourishment policy experiment. Second, we compare each case to the baseline to ensure that the mechanisms are working as we expect.

Scenarios Decreasing the Nourishment Subsidy

Low Alpha – base SLR

High Alpha – base SLR

Low Alpha - high SLR

High Alpha – high SLR

In the low alpha case, results are qualitatively the same as the main text findings. The difference is that property values can be sustained for longer because fewer high-income owners have entered the market at the onset of SLR. In the high alpha case, results are similar in that the subsidy only temporarily maintains property values in the face of SLR, but there is a shorter period in which high-income owners can enter before saturating the market. At first, the pattern of nourishment remains the same compared to the nourishment policy experiment in the main text. However, after roughly 30 years, an additional nourishment cycle begins, which the higher base property value justifies. So, we see some recovery in property values and beach width relative to the baseline 90% subsidy case.

Scenarios Compared to the Baseline

Low Alpha – base SLR

High Alpha – base SLR

Low Alpha - high SLR

High Alpha – high SLR

When we compare to the baseline, we can clearly see how the mechanism is working. Before SLR, the lower base property decreases property values, and the higher base property increases property values. Lower base property value decreases the average tax rate, and higher base property value increases the average tax rate. With lower alpha, the demographic shift narrows the gap in property value compared to the baseline, whereas with higher alpha, the demographic shift exacerbates the gap. Higher SLR just erodes property values overall relative to the baseline with lower SLR.

4. *Confidence Interval Inclusion: To enhance the informativeness of the results, it is recommended that the C-HOM model incorporates confidence intervals. This addition would provide a more comprehensive understanding of the statistical significance of the findings, thereby enhancing the robustness of the analysis.*

To this end, we repeatedly change the seed for the random number generator and re-run the baseline model. We use the resulting set of simulations to trace out 95% confidence intervals on the simulations. Supplemental Fig 8 shows the results. For most variables, the resulting confidence intervals are quite narrow. The main conclusions of the model are unchanged.

These suggestions aim to enhance the paper's overall clarity, validity, and applicability, contributing to a more comprehensive exploration of coastal real estate market dynamics.

Thank you again for these comments.

Reviewer #3 (Remarks to the Author):

This manuscript presents the Coastal Home Ownership Model, which uses an agent-based modeling approach to simulate the feedbacks in the coupled human-natural system in coastal communities experiencing emerging flood exposure due to sea level rise. The model is applied to examine how owners, renters, and investors value coastal property and invest in coastal management under various policy scenarios. Results indicate that certain management policies, such as providing subsidies for beach nourishment and tax advantages for high-income owners, will dampen and delay the effect of sea level rise on property values in coastal communities. Removing subsidies will allow property values to more accurately reflect the risk due to sea level rise but will also lead to a transition to wealthier ownership along the coast, pushing out lower-income owners. This highlights an interesting trade-off in the management of coastal property markets, with important economic and equity implications in how coastal communities respond to sea level rise. Overall, the manuscript is well-written, clearly organized, and addresses a topic of current interest to academics and practitioners across multiple fields. There are a few points of clarification that should be addressed before recommending the article for publication.

We thank the reviewer for providing constructive comments. Below we repeat the reviewer's comments in *italics* with our responses in plain text. We believe that addressing these comments has strengthened the clarity and the contribution of the paper.

Line numbers were not provided, so I will do my best to be clear about the relevant locations in the text.

We added line numbers for the revision.

Major comments:

- *The introduction should include some review of the literature on coupled human-natural systems modeling and clearly distinguish what advances are achieved through C-HOM. This would enable the reader to better understand the novelty of the work.*

Although a comprehensive review is beyond the scope of our paper, we add the following three paragraphs. The first is a general review of the coupled human-natural systems literature with a focus on land use change. The second is specifically about the work on coupled coastal systems and what C-HOM adds specifically to this literature. The third is parallel work on causality and coupled systems and why models like C-HOM are important for evaluating and the empirical literature and motivating future empirical work. We also include all of the references below these paragraphs.

This work contributes to advancing the growing literature on coupled human and natural systems. Humans are constantly changing the natural environment surrounding them and understanding dynamic feedback between human behavior natural systems often requires more than just superimposing an economic model on the physical or biological system (Liu et al. 2007). Applications of coupled modeling of dynamic human-natural feedbacks dates back at least to the 1960s when bioeconomic models were used to study the human and natural components of fisheries (Smith 1969; Abbott, Sanchirico, and Smith 2018). The literature expanded dramatically when researchers began using spatially explicit data to study land use and land cover change, urbanization patterns, and to evaluate conservation interventions (Plantinga et al. 1999; Carrion-Flores and Irwin 2004;

Polasky et al. 2008; An 2012; Lawler et al. 2014; Plantinga 2015). Progress in understanding coupled human-natural systems adds complexity by modeling non-linear feedbacks between physical processes and human responses across space and time (Liu et al. 2007; Levin et al. 2012).

In coastal systems, the evolution of the coastal-economic zone cannot be understood with methods in economics or coastal modeling alone; rather, it depends on complex interactions between physical coastal systems and economic behavior (McNamara and Werner 2008; McNamara et al. 2015). In these systems, incorporating relatively simple models of human behavior with a detailed geophysical model of coastal evolution (McNamara, Murray, and Smith 2011; Lazarus et al. 2018) and coupling simplified dynamics of coastal change with detailed economic decision-making (Smith et al. 2009; Gopalakrishnan et al. 2017) can generate new insights and emergent patterns in the coupled system (Murray et al. 2013). Adding complexity in any one dimension can reveal system characteristics that may not be consistent with simpler or more complex models. We add to this literature by endogenizing real estate values and demographic changes as functions of SLR risk in a model that also includes model features and couplings from this previous work, namely beach erosion, storm risk, the effects of beach width on property value, and local public finance decisions to rebuild beaches.

In the absence of modeling of a coupled human-natural system, researchers can also misinterpret empirical results and potentially draw the wrong policy implications (Smith 2014; Ferraro, Sanchirico, and Smith 2019; Schlüter et al. 2023). Even in simple models of coupled systems, state variables behave in non-intuitive ways such as being positively correlated over some time intervals and negatively correlated over others (Abbott, Sanchirico, and Smith 2018). As such, there is a growing need to use modeling to evaluate the reliability and plausibility of empirical evidence for causal claims and to elucidate potential mechanisms for surprising empirical findings (Ferraro, Sanchirico, and Smith 2019; Schlüter et al. 2023). The use of coupled systems modeling to inform empirical specifications can also lead to substantially different estimates, such as a value of beach that more than double the estimate that ignores the coupling (Gopalakrishnan et al. 2011).

Abbott, J.K., Sanchirico, J.N. and Smith, M.D., 2018. Common property resources and the dynamics of overexploitation: The case of the north pacific fur seal—A 42-Year Legacy. *Marine Resource Economics*, 33(3), pp.209-212.

An, L., 2012. Modeling human decisions in coupled human and natural systems: Review of agent-based models. *Ecological modelling*, 229, pp.25-36.

Carrion-Flores, C, Irwin E.G. 2004. Determinants of residential land use conversion and sprawl at the rural urban fringe. *American Journal of Agricultural Economics* 86(4):889–904

Ferraro, P.J., Sanchirico, J.N. and Smith, M.D., 2019. Causal inference in coupled human and natural systems. *Proceedings of the National Academy of Sciences*, 116(12), pp.5311-5318.

Gopalakrishnan, S., Smith, M.D., Slott, J.M. and Murray, A.B., 2011. The value of disappearing beaches: A hedonic pricing model with endogenous beach width. *Journal of Environmental Economics and Management*, 61(3), pp.297-310.

Lawler, J.J., Lewis, D.J., Nelson, E., Plantinga, A.J., Polasky, S., Withey, J.C., Helmers, D.P., Martinuzzi, S., Pennington, D. and Radeloff, V.C., 2014. Projected land-use change impacts on ecosystem services in the United States. *Proceedings of the National Academy of Sciences*, 111(20), pp.7492-7497.

Levin, S., T. Xepapadeas, A.-S. Crépin, J. Norberg, A. de Zeeuw, C. Folke, T. Hughes, et al. 2013. Socioecological systems as complex adaptive systems: Modeling and policy implications. *Environment and Development Economics* 18 (2): 111–32.

Liu, J., Dietz, T., Carpenter, S.R., Alberti, M., Folke, C., Moran, E., Pell, A.N., Deadman, P., Kratz, T., Lubchenco, J. and Ostrom, E., 2007. Complexity of coupled human and natural systems. *Science*, 317(5844), pp.1513-1516.

Murray, A.B., Gopalakrishnan, S., McNamara, D.E. and Smith, M.D., 2013. Progress in coupling models of human and coastal landscape change. *Computers & geosciences*, 53, pp.30-38.

Plantinga A.J., Mauldin T., Miller D.J. 1999. An econometric analysis of the costs of sequestering carbon in forests. *American Journal of Agricultural Economics*. 81:812–24

Plantinga, A.J., 2015. Integrating economic land-use and biophysical models. *Annual Review of Resource Economics*, 7(1), pp.233-249.

Polasky S., Nelson E., Camm J., Csuti B., Fackler P., et al. 2008. Where to put things? Spatial land management to sustain biodiversity and economic production. *Biological Conservation*. 141(6):1505–24

Schlüter, M., Brelsford, C., Ferraro, P.J., Orach, K., Qiu, M. and Smith, M.D., 2023. Unraveling complex causal processes that affect sustainability requires more integration between empirical and modeling approaches. *Proceedings of the National Academy of Sciences*, 120(41), p.e2215676120.

Smith, M.D., 2014. Fauna in decline: Management risks. *Science*, 346(6211), pp.819-819.

Smith, V.L., 1969. On models of commercial fishing. *Journal of Political Economy*, 77(2), pp.181-198.

• *The manuscript presents three scenarios that explore how changes from the baseline will influence the feedbacks observed in the system (page 6, paragraph 1). However, no rationale was provided to support the choice of the values for the beach nourishment subsidy and the timing/magnitude of outside market appreciation/depreciation. I think the paper would benefit from a sensitivity analysis that explores how changes in the choice of these values influences the system trajectory.*

Below we provide justification for each of our scenarios with references. The references are a mix of academic papers, grey literature, and news articles about particular places. We now include this information in the supplemental. We also conduct a sensitivity analysis as the reviewer suggests, and our qualitative results are unchanged. Below is a heatmap to summarize some of the sensitivity analyses, looking specifically at the effects of varying parameter values on property values over time. See the Supplemental Figures for more details that explore changes in the baseline, policy experiments, and variations of our scenarios.

Justification of the Baseline Case – 90% Nourishment Subsidy

The 90% baseline captures the typical case. Although there is considerable variation in how beach nourishment projects are funded, local funding from property taxes typically constitutes a small share of the total. For most beach nourishment projects, the federal subsidy has been approximately two thirds of the cost with the remaining one third financed by a combination of indirect federal subsidies for inlet stabilization and dredge disposal, state contributions, hotel taxes, local sales taxes, and local property taxes (NC DENR 2011; Brockbank et al. 2020; Gopalakrishnan, Landry, and Smith 2018; Star News 2021). Local property taxes are the non-subsidized component in the sense that it is paid directly by property owner beneficiaries of the project. Some projects are cost-shared between state, federal, and local funding with others having federal and state cost sharing that covers the entire cost of the project, e.g. in Louisiana (Elko et al. 2021). In some places, such as Kure Beach, NC and Wrightsville Beach, NC, local property taxes do not pay any share of the project (Town of Kure Beach 2019; Town of Wrightsville Beach 2023). Federal, state, and local funding shares vary substantially across projects in South Carolina with local funding providing no contributions in many instances but shouldering the entire burden in others (Houston 2021). Historically, between 65% and 85% of beach nourishment projects have had a federal component (Trembanis, Pilkey, and Valverde 1999). In Figure 4 of Valverde, Trembanis, and Pilkey (1999), 43% is federal storm and erosion, 14% is federal navigation, 6% is federal emergency, 2% is state, 18% is state and local cooperative agreements, and only 9% of funding is classified as local/private with 8 % as unknown.

Justification for Scenario 1 – Nourishment Subsidy Cut from 90% to 50%

The reduction to a 50% subsidy is a plausible change based on the political economy over the past two decades during which there has been momentum to reduce the federal share of funding for beach nourishment dramatically, decrease state contributions, and increase the share shouldered by local sources. Both the William Clinton and George W. Bush Administrations proposed cutting the two-thirds federal share in half, although Congress maintained subsidies at a higher level during their administrations (Tampa Bay Times 2019; CentralJersey.com 2001). Cutting the federal share in half alone would reduce the total subsidy to 57%, and there appears to be a similar push to force local communities to shoulder a greater share at the state level (Mullin, Smith, and McNamara 2019; Star News 2021).

Justification for Scenario 2 - Appreciation in outside real estate markets, namely a doubling in 50 years.

This scenario is based on historic real (inflation-adjusted) appreciation in national real estate markets in the United States and projecting that this appreciation will continue into the future. Specifically, doubling real prices in 50 years is a conservative projection of continued real estate appreciation from 1987 to the present (the period for which a consistent national real estate index is available). Doubling in 50 years implies a 1.4% appreciation rate. The historical real rate of appreciation in U.S. real estate markets (after adjusting for inflation) is 1.6%. This rate is calculated from the S&P/Case-Shiller U.S. National Home Price Index, Index Jan 2000=100, Monthly, Seasonally Adjusted, which is converted from nominal to real using the Consumer Price Index for All Urban Consumers: All Items in U.S. City Average, Index 1982-1984=100, Monthly, Seasonally Adjusted. The resulting 1.6% is the compounded real rate of appreciation over the period spanning January 1987 through July 2023.

Justification for Scenario 3 – Constant outside real estate markets and then dramatic decline, namely 90% depreciation in 50 years.

This scenario is exploratory in nature because real estate markets are not guaranteed to appreciate and could depreciate. We chose a substantial long-term depreciation rate to explore specifically if the demographic changes predicted by the model could be reversed. That said, this real rate of depreciation is not outside the rate of depreciation experienced recently on the decadal scale in housing markets. A 90% depreciation over 50 years corresponds to an annual rate of 4.5% depreciation. Based on the S&P/Case-Shiller U.S. National Home Price Index discussed above, between 2006 and 2012, housing markets depreciated at an annual rate of 7%, which is considerably faster than the 4.5% implied by our model.

Brockbank et al. 2020. Local Funding For Coastal Projects: An overview of practices, policies, and considerations. https://asbpa.org/wp-content/uploads/2020/01/Local-Funding-Report_Final_1.22.20.pdf

Elko, N., Briggs, T.R., Benedet, L., Robertson, Q., Thomson, G., Webb, B.M. and Garvey, K., 2021. A century of US beach nourishment. *Ocean & Coastal Management*, 199, p.105406.

Gopalakrishnan, S., Landry, C.E. and Smith, M.D., 2018. Climate change adaptation in coastal environments: modeling challenges for resource and environmental economists. *Review of environmental economics and policy*.

Houston, J.R., 2021. The economic value of beach nourishment in South Carolina. *Shore and Beach*, 89(3), pp.3-12.

Mullin, M., Smith, M.D. and McNamara, D.E., 2019. Paying to save the beach: effects of local finance decisions on coastal management. *Climatic Change*, 152, pp.275-289.

NC DENR. 2011. NC Beach and Inlet Management Plan – Final Report

<https://www.deq.nc.gov/documents/pdf/bimp/bimp-section-xii-funding-prioritization-formatted/download>

<https://www.starnewsonline.com/story/news/2021/08/13/north-carolina-beach-maintenance-costs-rising-amid-climate-change-rising-seas/5567860001/>

<https://www.townofkurebeach.org/sites/default/files/uploads/beach-nourishment-rot-brochure-2019.pdf>

<https://archive.centraljersey.com/2001/07/05/beach-replenishment-still-a-federal-project-presidents-plan-to-reduce-funding-overturned-in-the-house-of-representatives/>

<https://www.tampabay.com/archive/1999/07/06/costly-beach-proposal-resisted/>

Town of Wrightsville Beach, NC Beach Management Plan. 2023. Draft Report, August 14, 2023

Trembanis, A.C., Pilkey, O.H. and Valverde, H.R., 1999. Comparison of beach nourishment along the US Atlantic, Great Lakes, Gulf of Mexico, and New England shorelines. *Coastal Management*, 27(4), pp.329-340.

Valverde, H.R., Trembanis, A.C. and Pilkey, O.H., 1999. Summary of beach nourishment episodes on the US east coast barrier islands. *Journal of Coastal Research*, pp.1100-1118.

- *Certain modeling components should be explained in greater detail to enable the reader to evaluate the approach and understand potential limitations. For example:*

- o How is the volume of sand (and thus the price of nourishment) determined over time?*

The literature suggests sand costs are about \$10 per cubic yard (Gopalakrishnan, Landry, and Smith 2018; Elko et al. 2021; Cutler, Albert, and White 2020). There are also fixed costs for planning and mobilizing equipment that are included in the costs. Sand volume is based on the alongshore length and width of the project (Smith et al. 2009; McNamara et al. 2011). These parameters are now included and documented in the parameter table. The revised parameter table includes descriptions and justifications for the other parameters as well.

Cutler, E. M., Albert, M. R., & White, K. D. (2020). Tradeoffs between beach nourishment and managed retreat: Insights from dynamic programming for climate adaptation decisions. *Environmental Modelling & Software*, 125, 104603.

Elko, N., Briggs, T.R., Benedet, L., Robertson, Q., Thomson, G., Webb, B.M. and Garvey, K., 2021. A century of US beach nourishment. *Ocean & Coastal Management*, 199, p.105406.

Gopalakrishnan, S., Landry, C.E. and Smith, M.D., 2018. Climate change adaptation in coastal environments: modeling challenges for resource and environmental economists. *Review of environmental economics and policy*.

Smith, M.D., Slott, J.M., McNamara, D. and Murray, A.B., 2009. Beach nourishment as a dynamic capital accumulation problem. *Journal of Environmental Economics and Management*, 58(1), pp.58-71.

- o Does beach nourishment have any effect on the level of exposure (i.e., can a wider beach reduce some flooding impact)? Or is beach nourishment only viewed from an amenity perspective? If the latter is true, which I believe is the case, is it possible to incorporate hazard reduction due to nourishment in the risk premium formulation?*

We chose not to add this feature because the empirical literature is somewhat ambiguous on the mechanism, and the risk terms in our model are already complicated and not easily parameterized. Quasi-experimental studies that exploit spatial and temporal differences in adaptation investments and the occurrence of hazard events show that nearshore housing prices capitalize potential storm risk reduction from beach nourishment in North Carolina (Qiu & Gopalakrishnan, 2018) and the construction

of vegetated dunes along the New Jersey coast (Dundas, 2017). Both of these are already cited in the paper. However, empirical analyses that separate the benefits of erosion control and hazard mitigation are limited by the reality that nourishment often is done in conjunction with dune construction.

Qiu, Y., & Gopalakrishnan, S. (2018). Shoreline defense against climate change and capitalized impact of beach nourishment. *Journal of Environmental Economics and Management*, 92, 134–147.

Dundas, S. J. (2017). Benefits and ancillary costs of natural infrastructure: Evidence from the New Jersey coast. *Journal of Environmental Economics and Management*, 85, 62–80.

o A nourished dune height is listed in Table S1, but I don't recall this being defined in the model. Please clarify if dune height is considered in the model.

We removed this parameter from the table. It was an artifact of an earlier draft of the model.

o Please provide an explanation of how the scaling parameters (a_1 and a_2) were determined.

Detailed parameter explanations and documentation are now included in the notes to Table S1. The table also provides motivation for our sensitivity analyses.

Minor comments:

• *Page 4, paragraph 1: Please provide a definition for a “bubble” in the context of this paper. While this term is widely used, I think a clear explanation would be helpful for the general reader.*

We add this definition: “A bubble occurs when prices rise substantially above expected prices based on market fundamentals over a short period of time.”

• *Page 4, paragraph 5: Why is a 150-year time horizon chosen for this analysis?*

We add the following text: “This length of time allows us to consider longer horizons than a typical 30-year mortgage; run the model for 50 years without SLR as a initial period to understand internal mechanisms in the model; and evaluate large, long-term effects of SLR and changing storm climate (over the subsequent 100 years).”

• *Page 6, paragraph 2: Is the “barrier height” a reference to the barrier island itself? Or is there assumed to be another structural barrier providing flood protection? Please clarify.*

Yes, the elevation of the barrier island. In the text, we added “i.e., elevation of the barrier island”

• *Page 17, paragraph 1: How is the spatial extent of the nourishment unit determined? For this hypothetical case, why is it important that it be “smaller than a town but larger than a census block group”?*

Typically towns decide on nourishment projects if they are partly funded through property taxes. However, many towns throughout the U.S. have special municipal tax districts that face different tax rates to fund specific services. This is a common practice for beach nourishment. We add the following to the text.

This spatial extent allows for the decision-making unit for a nourishment project that is funded by property taxes to be a town, but it includes the possibilities that the town may not nourish its entire beach and that there are special tax districts within the town that pay higher rates to fund the project (Mullin, Smith, and McNamara 2019).

- *Page 18, first line: Include “equation” before the number 4.*

Done.

- *Page 20, equation 13: Please define all terms that are not previously defined. I think a_3 should be a_2 , as referenced in Table S1.*

That’s correct. We add to the text here, “where $\$MSL_t$ is mean sea level, which changes with SLR, and $\$h^{elev}$ is initial barrier elevation.” We add more clarifications in Table S1.

Thank you again for these comments.

REVIEWERS' COMMENTS

Reviewer #2 (Remarks to the Author):

After the revision, I think the paper has improved significantly and is ready for publication. Most of my comments in the first round of review are addressed.

Reviewer #3 (Remarks to the Author):

The authors have adequately addressed all of the reviewer comments and have provided detailed revisions in the new version of the manuscript. I support publication of the manuscript in its revised version.